



# Nutrients Dissolution Kinetics of Aerosols at Qianliyan Island, the Yellow Sea by a High Time-resolution Nutrient Dissolution Experiment, Potential Linkages with Inorganic Compositions and P solubility controlled factors

Ke Zhang[1,2], Lijun Han[1,3], Sumei Liu[1,2], Lingyan Wang[1,2]

[1] Key Laboratory of Marine Chemistry Theory and Technology MOE/College of Chemistry and Chemical Engineering, Ocean University of China, Qingdao 266100, China

[2] Laboratory for Marine Ecology and Environmental Science, Qingdao National Laboratory for Marine Science and Technology, Qingdao 266237, China

[3] Present at Hebei Province Environmental Monitoring Center

**Correspondence:** Sumei Liu (sumeiliu@ouc.edu.cn)

**Graphic abstract:**

## Nutrient release of aerosol particles in diverse water conditions

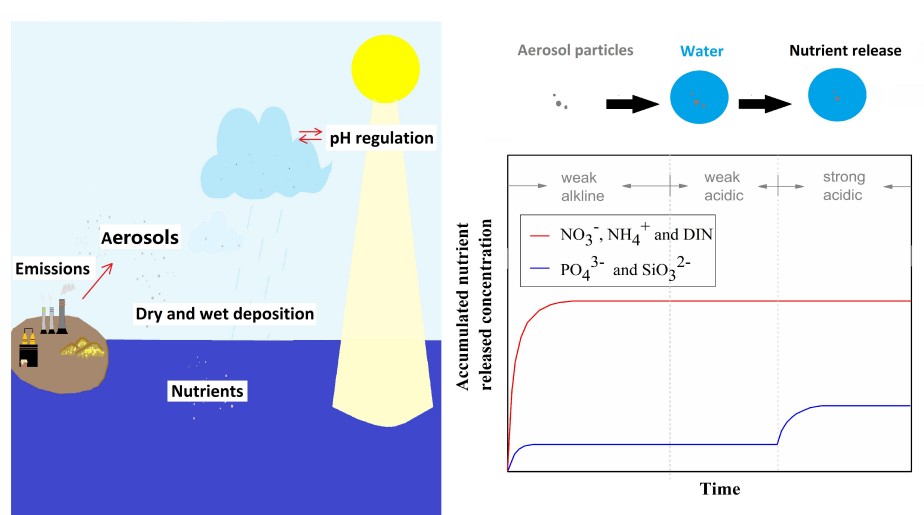

**Abstract.** A series of high time-resolution nutrient dissolution experiments were designed to determine the soluble fraction of atmospheric nutrients and reveal the short-time dissolution processes, patterns and kinetics of nutrient elements in aerosols. Aerosols that represented an important part of atmospheric transport path over the East Asian to West Pacific were leached by Milli-Q water and aged seawater at gradient pHs for certain time duration. Varied nutrient dissolution curves indicated that





aerosol inorganic N, P and Si species dissolution reactions were quasi-first-order. Particularly, prominent factors influenced P solubility were source and acidity. Ratios of acid-soluble to water-soluble nutrient concentrations in high time-resolution dissolution experiments and ultra-sound extractions were 1.0 (0.9-1.1) for $NH_4^+$ and $NO_3^-$, 2.4 (2.1-2.6) for $PO_4^{3-}$ and 2.5 for $SiO_3^{2-}$,
demonstrating that inorganic N species were inclined to immediate and complete dissolution due to fine
particles formed by gas-particle transformation, inorganic P (Fe-P, Ca-P and De-P) and Si were tended to dissolve more in strong acidity mainly because of coarse soil-derived mineral particles. Compared with the slow dissolution of inorganic P and Si, the rapid dissolution of inorganic N can affect the composition of marine nutrients and marine primary productivity.

## 1.   Introduction

Atmospheric deposition supplies dissolved or particulate nutrients onto the surface ocean, which is a vital source of nutrient transportation into ocean (Duce et al., 2008; Guieu et al., 2010). Also, atmospheric deposition enhances marine primary production, showing potential linkages with phytoplankton growth and their community structures (Doney, 2010; Anderson et al., 2011; Nenes et al., 2011; Tréguer et al., 2013; Guieu et al., 2015; Siswanto, 2015).

For both marginal seas and oceans, atmospheric deposition is of great significance, so atmospheric deposition flux is wildly proposed to assess aerosol nutrient levels. The molar nutrient concentrations in aerosols used for calculations of atmospheric deposition dry flux are extracted by diverse methods. The prevailing extraction method was ultrasonic extraction method using Milli-Q water with different time in-length (0.5-1h) at 0℃ or at room temperature (Cornell et al., 1995; Nakamura et al., 2005; Shi et al.,
2010; Qi et al., 2017). Except for ultrasound methods, a mini-loop circulation system was invented to measure aerosol soluble fraction (Eyckmans et al., 2001). However, these extraction conditions do not directly reproduce those under which aerosol components would be released into seawater. Few studies have showed the concern about the comparisons between pure-water and seawater-soluble nutrient



concentrations of aerosols since last two decades. For major N species, no significant differences were

observed between pure-water and seawater extractions (Chen et al., 2006; Kocak, 2015). It was a pity

that results for soluble P in pure-water and seawater were controversial (Markaki et al., 2003; Chen et

al., 2006; Kocak, 2015). Study of Markaki et al. (2003) indicated no statistical difference, on the

contrary, studies of Chen et al. (2006) and Kocak (2015) revealed that dissolution of $PO_4^{3-}$ in seawater

were 11% and 52% lower than that in pure water. For soluble Si, Kocak (2015) showed that dissolution

of $SiO_3^{2-}$ in seawater was 66% lower than that in pure water.

Aerosol soluble P and Si data were scarce compared to soluble N species, so researches focused on P

and Si solubility under the assumption that phytoplankton prefers soluble nutrients. Regardless of

different extractions, P solubility of Atlantic ocean aerosol, Saharan and Asian dust aerosols were 32%

(0.01-87%), 8% (2.3–67%) and 37% (9.3-90%), respectively (Baker et al., 2006a; Baker et al., 2006b;

Wang et al.,2017). Si solubility of Saharan aerosol and Asian dust aerosol ranged 0.12% (0.02-1.1%)

and 0.16% (0.02-0.74%) (Baker et al., 2006b; Wang et al.,2017). These assessments of solubility gave a

general view of solubilisation, contributing to simple biogeochemical model (Harrison, 2000;

Krishnamurthy et al., 2010), arising attentions on aerosol nutrient accretions via atmospheric transfer

path (Anderson et al., 2010; Nenes et al., 2011). Nevertheless, aforementioned investigations did not

further dig into dissolution kinetics, so more information could be given when considered dissolved

nutrient concentrations as a function of time and pH. Long-time (72h) laboratory leaching experiments

tested aerosol P dissolution kinetic as a function of time, showing 2-9 fold soluble P increase after 72h

compared with initial soluble P concentration (Mackey et al., 2012). Besides, long-time (48h) batch

dissolution experiments and short-time (60s) step-by-step dissolution experiments that setting a series of

descending pH gradients, of which both using aerosol precursors that tested P dissolution kinetic as a

single variable of pH, could quantified the $H^+$ concentration threshold value that aerosol precursors

started dissolved and soluble P increased 6 times (Aghnatios et al., 2014; Stockdale et al., 2016).

Moreover, short-time (75min) nutrient dissolution experiments using modified mini-loop system that



monitored dissolution process in greater detail via high time-resolution were presented acid promotion, solubility and successive observational results, which could make dissolution process be modeled and give the dissolution rate, dissolution equation information (Wang et al.,2017). However, these dissolution kinetic studies wholly used Milli-Q water as leaching solution, making the scenario of seawater dissolution kinetics of aerosols virtually unknown.

A strong gradient of heavy atmospheric N depositions from East Asian to light atmospheric N
depositions in the central Pacific Ocean had been reported (Zhang et al., 2011). Our undeveloped sampling site, Qianliyan island, was on the atmospheric transport path over the East Asian to West Pacific, so it was an excellent location for aerosol collection and regional study. In this study, we aimed to investigate the magnitude of aerosol nutrient concentrations, the pattern of aerosol nutrients dissolution and controls of P solubility of aerosols, of which particular interest is the design to model
the dissolution curves that providing details about short-time dissolution kinetics. Experiments were carried out using aerosols collected at Qianliyan island to mimic aerosol dissolution process in natural water (sea water and atmospheric water) undergone different pH conditions in the face of soaring anthropogenic emissions and climate change.

## 2.  Material and Methods

### 2.1.  Site Description, Sample Collection and Sample Selection

Qianliyan island (36°16′N, 120°23′E), as a uninhabited sampling site over the north-west Yellow Sea, was under downwind area of heavy anthropogenic emissions in winter (Liu et al., 2012) and Asian dust transport in spring and summer (Zhang et al., 2003; Uno et al., 2009; Gao et al., 2014; Tao et al., 1987; Huang et al., 2007). Avoided unnecessary interference from nearby pollution sources, this island was an
excellent sampling site, but the hard conditions on the island made it difficult to get samples.

Aerosols were weekly collected from January to July in 2011 and from March to October in 2012 at the top of Qianliyan Island (93.5 meters above sea level) by a high-volume sampler (KC-1000,



LAOSHAN MOUNTAIN ELECTRONIC INSTRUMENT FACTORY CO., LTD.). The discontinuity of aerosol collection from August, 2011 to February, 2012 was caused by power supply problem.Total

suspended particulate (TSP) samples were collected using standard poly-carbonate filters in 2011 and Whatman 41 cellulose fiber filters in 2012. Both filters are reported being suitable for TSP collection and chemical analysis. These filters were acidic cleaned by hydro-chloride acid solution ($0.01mol \cdot L^{-1}$) and washed acid out until the same pH as Milli-Q water. Then pretreated filters were clearfully loaded onto the filter holder under clean condition before sample collection. The flow rate was set $1.05m^3 \cdot min^{-1}$

and total sampling time was 20h. Totally, 39 aerosols were collected.

      Sample Selection was based on season and main source. In order to reduce the uncertainty of the variable direction of the air mass, all backward trajectories that computed by the Hysplist4 software and integrated by the MeteoInfoMap software (Fig. 1) were based on the airborne backward trajectory of 72h in the intervals of an hour at the height of 1000 m, the lower height of marine atmospheric

boundary layer (MABL). North-west (NW) was prevailing source direction in spring, autumn and winter (Fig. S1), which was consistent with the typical Asian monsoon transport direction. South-west (SW) was the second major source direction in spring with the characteristics of anthropogenic emissions in southern China. Therefore, six aerosols (spr-SW, spr-NW, sum-NW, aut-NW, win-NW1 and win-NW2) capturing seasonal main features were selected for high time-resolution dissolution

experiments. From spring to early summer, enormous quantity of mineral dust particles caused by East Asian dust storms was lifted from arid and semi-arid regions in Northeast Asia and transported a long distance. As showed in Fig. 1, spr-NW aerosol was affected by an intense sandstorm from arid and semi-arid regions of Inner Mongolia and northern China (China Meteorological Administration, 2011) with high TSP mass concentration (Table S1); while other aerosols had not been reported to be affected

by sand-storms, among them, spr-SW aerosol was mainly from Jiangsu and Anhui provinces and transported over Shandong province, sum-NW and aut-NW aerosols were from Russia and transported through west-north China and both win-NW1 and win-NW2 aerosols were originated from the Inner



Mongolia province.

## 2.2. Ultra-sound Extraction and Chemical Analysis

Aerosol sub-sample (one 47mm diameter circle piece cut off from sampling filter) was extracted with Milli-Q water or a HCl solution (1mol·L$^{-1}$) for an hour at room temperature and then extraction was filtered through 0.45μm PES membrane. The nutrient concentrations of filtered extractions were measured by nutrient analyzer (SKALAR SAN$^{plus}$, Skalar Analytical B.V.). The detection limits of $NH_4^+$, $NO_3^-$, $PO_4^{3-}$ and $SiO_3^{2-}$ were 0.23 μmol·L$^{-1}$, 0.20 μmol·L$^{-1}$, 0.01 μmol·L$^{-1}$ and 0.10 μmol·L$^{-1}$,

respectively. The detection precision of $NH_4^+$, $NO_3^-$, $PO_4^{3-}$ and $SiO_3^{2-}$ were 2.6%, 2.2%, 1.2% and 1.4%, respectively.

## 2.3. High Time-resolution Dissolution Experiment

    High time-resolution dissolution experiment used a small-volume flow-through leaching system (Fig. 2) based on Eyckmans et al. (2001) with a non-closed loop. The device was exactly the same as the one

used by Wang et al. (2017), but the experimental conditions that specifically referring to leaching solution types and pHs were different. To simulate the natural plots, aged sea water and Milli-Q water were used as analogs of surface seawater and atmospheric water, respectively. Under these natural plots, pH gradient from 7.8 to 5.5 to 2.0 were used to simulate the deterioration of environmental conditions. Specifically, Milli-Q water at pH 7.8, 5.5 and 2.0 simulated weak alkaline atmospheric water,

unpolluted natural atmospheric water and extreme acidic atmospheric water, respectively. Atmospheric water like rainwater could be weak alkaline due to the presence of alkaline elements, such as Ca and Mg, and weak acidic because of the equilibrium of water and $CO_2$, but could not be strong acidic that having pH lower than 3 (Losno et al., 1991; Chen et al., 2006); however, atmospheric water like cloud water may reach such low pH value owing to high acid/water ratio and eastern Asian cloud water had been

reached 1 (Meskhidze et al., 2003). Aged seawater at pH 7.8, 5.5, 2.0 simulated natural surface sea water, acidic seawater and extreme acidic seawater, respectively. The acidic seawater scenarios were



unlikely to occur due to the buffering capacity of seawater when facing current oceanic acidification, but they did show the scenarios when oceanic acidification exceeded the marine buffering capacity and/or the effect of diverse ionic strength (IS) on nutrient dissolution. At the same time, high time-resolution dissolution experiment required a reasonable aerosol area and leaching volumes for the simulation of the atmospheric deposition over Yellow Sea. The surface SPM concentrations of the Yellow Sea were characterized by typically ranging from 1 to 10 $mg \cdot L^{-1}$ in the central areas and up to 200 $mg \cdot L^{-1}$ in coastal areas (Park et al., 2001; Dobrynin et al., 2009; Zhang et al., 2018). The particulate concentrations that total volume (281.25mL) of leaching solution divided aerosol particle mass of each 17.3$cm^2$ sub-samples in leaching system were ranged from 4.9 $mg \cdot L^{-1}$ to 34.8 $mg \cdot L^{-1}$. Experimental simulated particular concentrations were lower than surface SPM concentrations of the Yellow Sea, so our settings of sample area, leaching speed and time were reasonable.

In our leaching system, aerosol sub-sample was put into a mini-volume container, leaching liquid was pumped through at a constant velocity of 2.5 $mL \cdot min^{-1}$ and disposal filter was set at the end of the device to keep particles excluded. Leaching liquid was either Milli-Q water or aged seawater with pH gradient from 7.8 to 5.5 to 2.0. Basic or acid condition was adjusted by NaOH or HCl solution. Leachates were collected at a 90s interval. The leaching time periods that 45 min for pH 7.8, 30 min for pH 5.5 and 37.5 min for pH 2.0, could ensure leaching system had reached the dissolution balance. The nutrient concentrations in leachates were measured by nutrient analyzer. All operations were carried out at room temperature (20±5 °C). The sum of nutrient concentrations of all leachates in each aerosol from 0 to 75 minutes represented water-soluble nutrient concentrations and that from 0 to 112.5 minutes represented acid-soluble nutrient concentrations.

## 2.4. SEDEX Method

Modified SEDEX Method (Ruttenberg, 1992; Huerta-Diaz et al., 2005) was used for aerosol P form measurements. There were 4 steps ascribed as follows: Step 1: The sub-sample was firstly extracted 6h



by 15 mL citrate/dithionite/bicarbonate (CDB) mixture solution (0.22mol·L$^{-1}$ C$_6$H$_5$Na$_3$O$_7$+1mol·L$^{-1}$ NaHCO$_3$+0.033mol·L$^{-1}$ Na$_2$S$_2$O$_4$) at pH 7.6, then extracted 2h by 15mL MgCl$_2$ solution (1mol·L$^{-1}$) at pH 8 and 2h by 15mL Milli-Q water. And these extraction solutions were gathered together as one extraction. Step 2: The residue of sub-sample was extracted 2h by 15mL acetate buffer solution at pH 4

(1mol·L$^{-1}$ CH$_3$COONa buffered to pH 4 with acetate acid), then extracted 1h by 15mL MgCl$_2$ solution (1mol·L$^{-1}$) at pH 8 twice and 1h by 15mL Milli-Q water. And these extraction solutions were gathered together as one extraction. Step 3: The residue of sub-sample of step 2 was extracted 16h by 19.5mL HCl solution (1mol·L$^{-1}$). Step 4: The residue of sub-sample of step 3 was ashed 2h at 550°C and then extracted 24h by 19.5mL HCl solution (1mol·L$^{-1}$). After finished entire four steps, the phosphorus

concentrations were measured by nutrient analyzer. The forms of P in each step were defined as Fe-bound P (including exchangeable P, Fe-P), Ca-bound P (Ca-P), debris P (De-P) and organic P (OP), respectively. The sum of these four P forms was total P(TP). TP concentration of soil standard GBW07314 was 20.11 ± 0.33 μmol/g (n=7), the standard value was 20.85 ± 1.97 μmol/g and the recovery was 97%. The measured precisions of Fe-P, Ca-P, De-P, OP and TP in soil standard were 2.4%,

1.8%, 0.95%, 4.7% and 1.3%, respectively.

**2.5.  P Solubility**

P solubility (s) was calculated as   $s = \dfrac{c_t}{c_{TP}}$ , where $c_t$ was the accumulated dissolved P concentration at any time t for high time-resolution dissolution experiment and $c_{TP}$ was the total P concentration measured by SEDEX Method; specifically, Milli-Q water-soluble P solubility was presented as   $s = \dfrac{c_{75}}{c_{TP}}$ ,

aged seawater-soluble P solubility was   $s = \dfrac{c_{45}}{c_{TP}}$ , Milli-Q acid-soluble and aged seawater acid-soluble P



solubility were $s = \dfrac{c_{112.5}}{c_{TP}}$ . These calculations were under the assumption that dissolved inorganic P in

weak basic condition (pH 7.8) could leach in weak acidic condition (pH 5.5) and dissolved inorganic P in weak acidic condition could leach in strong acidic condition (pH 2.0).

## 2.6. Modeling of Dissolution Curves

Modeling of dissolution curves for high time-resolution dissolution experiments were conducted as follows: when nutrient concentrations in leachates were exponential decreasing, accumulated dissolution curves were mathematically expressed as uniform equations that based on Avrami Model (Avram, 1939), which were $c_t = c_e * (1 - e^{-kt^n})$  $t \leq t_e$ and $c_t = c_e$  $t > t_e$  (Equations 1), where t was an arbitrary time, $t_e$ was the time reached dissolution equilibrium, $c_t$ was the accumulated concentration at time t, $c_e$ was the accumulated concentration at dissolution equilibrium, k was the dissolution rate constant and n was the order of reaction and when nutrient concentrations in the leachates were slightly agitated with time, the cumulative dissolution curve based on the zero-order kinetic model were simply expressed as $c_t = k * t$  (Equation 2), where t was an arbitrary time, k was the dissolution rate constant and $c_t$ was the accumulated concentration at time t. Accordingly, aerosol dissolution rate (r) were described as $r = c_e * k * n * e^{-kt^n} * t^{(n-1)}$ for non-liner accumulated dissolution curves and $r = k$ for liner accumulated dissolution curves.

## 3.  Results

### 3.1.  Aerosol Nutrient Concentrations and Water-/Acid-soluble Ratios

Milli-Q water-soluble nutrient concentrations for ultrasound extractions showed a large range, with arithmetic mean values of $154 \pm 91.6$ nmol·m$^{-3}$ for $NH_4^+$, $135 \pm 91.3$ nmol·m$^{-3}$ for $NO_3^-$, $0.51 \pm 0.30$ nmol·m$^{-3}$ for $PO_4^{3-}$ and $0.54 \pm 0.36$ nmol·m$^{-3}$ for $SiO_3^{2-}$, respectively. Milli-Q acid-soluble $NH_4^+$, $NO_3^-$

and $PO_4^{3-}$ concentrations for ultrasound extractions also showed a large range, with arithmetic mean values of $189\pm115$ nmol·m$^{-3}$ for $NH_4^+$, $135\pm81$ nmol·m$^{-3}$ for $NO_3^-$ and $1.3\pm1.1$ nmol·m$^{-3}$ for $PO_4^{3-}$,

respectively. Acid-/water-soluble ratios were 1.1 for $NH_4^+$ and 0.9 for $NO_3^-$ and 2.2 for $PO_4^{3-}$ (Fig. 4). In the high time-resolution dissolution experiments, varied Milli-Q/aged water-soluble nutrient concentrations were lower than Milli-Q/aged acid-soluble nutrient concentrations (Table S3). Acid-/water-soluble ratios were 1.0 for $NH_4^+$ and 1.0 for $NO_3^-$, 2.6 for $PO_4^{3-}$ and 2.5 for $SiO_3^{2-}$.

In both measurements, concentrations of acid-soluble nitrogen species were 0.9-1.1 times higher than water-soluble concentrations and concentrations of acid-soluble $PO_4^{3-}$ were 2.2-2.6 times higher than

water-soluble $PO_4^{3-}$ concentrations regardless of different leaching liquid, indicating that it was difficult for aerosols to release more inorganic nitrogen even if the dissolution time increased or the acidity increased, but the acid could increase P solubility.

### 3.2.  P Forms

Aerosol TP concentrations varied from 5.71 to 20.47 μmol·g$^{-1}$ and the highest TP concentration was

found in spr-NW aerosol (Fig. 3). Concentrations of total inorganic P (Fe-P, Ca-P and De-P) represented 90-100% of TP were varied between 5.6 and 19.0 μmol·g$^{-1}$. Fe-P concentrations were 4.27-7.02 μmol·g$^{-1}$, Ca-P concentrations were 1.31-11.34 μmol·g$^{-1}$ and De-P concentrations were 0-0.80 μmol·g$^{-1}$. The SEDEX extractions revealed that Ca-P was the dominant P forms in spr-NW aerosol, whereas Fe-P was the dominant P forms in other aerosols. In Asian sand-dust, percentage of Ca-P of TP ranged

48%-94% and that of Fe-P ranged 0.12%-14% (Yang, 2012). Spr-NW aerosol had typical composition of Asian sand-dust, containing 55% Ca-P but 33% Fe-P. Change in Fe-P proportion was possibly aroused by newly formed iron particles during cloud processing (Shi et al., 2009) and evinced that aerosol mightily had been experienced atmospheric acidification.

### 3.3.  Depictions and Equations for Dissolution Curves

Aerosol $NH_4^+$ and $NO_3^-$ had similar dissolution process, $PO_4^{3-}$ and $SiO_3^{2-}$ had similar dissolution



process (Fig. 5 & Fig. S2). Raw $NO_3^-$ and $NH_4^+$ concentrations were rapidly reduced from high initial concentrations during 0-6/7.5 min and followed by no longer release except for win-NW2 aerosol. However, $NH_4^+$ and $NO_3^-$ dissolution of win-NW2 aerosol during 45-75 min and 75-112.5 min were small, which were less important because 95% ± 5% of total accumulated $NH_4^+$ or $NO_3^-$ had been released at pH 7.8. Hence, accumulated $NH_4^+$ and $NO_3^-$ dissolution curves could be described as Equations 1. Raw $PO_4^{3-}$ and $SiO_3^{2-}$ concentrations were rapidly reduced from high initial concentrations to near-zero within 6 min, rapidly soared to the second high concentrations after 75 min and then reduced to varying degrees. Correspondingly, accumulated $PO_4^{3-}$ and $SiO_3^{2-}$ concentrations at pH 7.8 accounted for 33 ± 14% and 34 ± 21% of the total accumulated $PO_4^{3-}$ and $SiO_3^{2-}$ concentrations, respectively. After the first pH change, $PO_4^{3-}$ and $SiO_3^{2-}$ dissolution of aerosols were overall at a slow rate. At pH 5.5, accumulated $PO_4^{3-}$ and $SiO_3^{2-}$ concentrations accounted for 6 ± 3% and 13 ± 14% of the total accumulated $PO_4^{3-}$ and $SiO_3^{2-}$ concentrations, respectively. After the second pH change, $PO_4^{3-}$ and $SiO_3^{2-}$ dissolution of aerosols were from quick to slow until dissolution curves reached a plateau. At pH 2.0, accumulated $PO_4^{3-}$ and $SiO_3^{2-}$ concentrations accounted for 61 ± 16% and 52 ± 30% of the total accumulated $PO_4^{3-}$ and $SiO_3^{2-}$ concentrations. Therefore, aerosol accumulated $PO_4^{3-}$ and $SiO_3^{2-}$ dissolution curves could be depicted as three stages classified according to different dissolution processes at corresponding pH, so accumulated $PO_4^{3-}$ and $SiO_3^{2-}$ dissolution curves could be described as combinations of Equation 1 and Equation 2, of which Equation 1 was used to fit in curves at pH 7.8 and 2.0 and Equation 2 was used to fit in curves at pH 5.5.

Equation parameters for high time-resolution dissolution curves were given in Table 1 and Table 2. The dissolution rate constant (k) of $NH_4^+$ and $NO_3^-$ exceeded k of $PO_4^{3-}$ and $SiO_4^{3-}$, reflecting that aerosol N species dissolution was easier and faster. The values of k were ranged 0.0047-0.90 $min^{-1}$ for $PO_4^{3-}$ and 0.0005-0.33 $min^{-1}$ for $SiO_3^{2-}$ in this study and ranged 0.13-0.80 $min^{-1}$ for $PO_4^{3-}$ and 0.10-0.91 $min^{-1}$ for $SiO_3^{2-}$ in reported study (Wang et al., 2017). Similar k ranges for $PO_4^{3-}$ and $SiO_3^{2-}$ confirmed that aerosol P and Si dissolution were harder and slower than aerosol N species. However, mean k



values of Qianliyan aerosols were lower than mean k values of Qingdao aerosols. The experimental apparent reaction order (n) was expressed dissolution reaction mechanism that is the relationship between the dissolution reaction rates and nutrient concentrations. Mean n values were 0.72±0.42 for $NH_4^+$, 1.01±0.44 for $NO_3^-$, 0.80±0.26 for $PO_4^{3-}$, and 0.96±0.39 for $SiO_3^{2-}$, respectively. Therefore, the aerosol inorganic N, P and Si dissolution were quasi-first-order reactions. This was in consistence with reported n values of P and Si dissolution equations (Truesdale et al. 2005; Wang et al. 2017). Furthermore, P dissolution in aged seawater needed 37.5-42 min for equilibrium, whereas P dissolution in Milli-Q water needed within 10-15 min for equilibrium (Wang et. al., 2017), which meant aerosol P dissolution were different in atmosphere and in sea-water. Additionally, accumulated P/Si molar ratio curves (Fig. 6) that tended to be fixed over time were indicated the mix release of P and Si in aerosols that may caused by non-stoichiometric and/or stoichiometric dissolution of apatites and silicates.

### 3.4. P solubility

Phosphorus solubility varied within time and tended to stabilize in a short time (Fig. 7). Overall, the aged seawater-soluble P solubility of SW aerosol was greater than that of all NW aerosols and the aged seawater acid-soluble P solubility of SW aerosol was the second largest solubility of all aerosols, which revealed that P in SW aerosols may be attacked by acidic gas that released from the source and during air-masses long-distance transport and part of acid-soluble P was converted into seawater-soluble phosphorus. For same NW aerosols, aged seawater-soluble P solubility were 7%, 9%, 27% and 16±3% in spring, summer, autumn and winter, respectively; aged seawater acid-soluble P solubility were 41%, 18%, 54% and 69±35% in spring, summer, autumn and winter, respectively. The difference that seawater acid-soluble P solubility minus aged seawater-soluble P solubility was indicated the maximum dissolved P potential in seawater.

Milli-Q water-soluble P solubility of spr-NW and win-NW2 aerosols were both 8% and Milli-Q acid-soluble P solubility aerosol of them were 36% and 17%, respectively. Compared to P solubility in





Milli-Q water for Qianliyan aerosols, Milli-Q water-soluble P solubility of spr-NW and win-NW aerosols were 9% and 22%, respectively, and Milli-Q acid-soluble P solubility aerosol of them were 95% and 65%, respectively for Qingdao (a city adjacent to the Yellow sea) aerosols (Wang et. al., 2017). For same source aerosols, Milli-Q acid-soluble P solubility that represented the maximum potential ability of acid to modify aerosol P during atmosphere transport was higher in spring than in winter

though Milli-Q water-soluble P solubility was similar in spring and winter.

## 4. Discussion

### 4.1. Relevance of Aerosol Inorganic Components and Dissolution Patterns

The dissolution patterns were closely related to aerosol inorganic components because components determined structures and structures determined their natures of dissolution. Dissolution can be induced

by concentration diffusion and/or ions attraction effect, of which ions attraction effect referred to aerosol chemical reactions with hydroxyls ($OH^-$), water molecules ($H_2O$), protons ($H^+$) and/or organic ligands under corresponding basic, neutral and acid solutions. In high time-resolution dissolution experiments, the leachates were subsequently moved out of the reaction system in time series, so dissolution at each time interval was unsaturated. Considered that an average of 98% (96-100%) of the

cumulative dissolved $NH_4^+$ and $NO_3^-$ in aerosols was dissolved at pH 7.8, the dissolution of aerosol N species was mainly induced by existing nutrient concentration differences and the dissolution of aerosol N species would not stop until soluble components had been used up. The main deliquescent $NO_3^-$ and $NH_4^+$ components in aerosols, such as $NH_4Cl$, $Ca(NO_3)_2$, $KNO_3$, $Mg(NO_3)_2$, $NaNO_3$, $NH_4NO_3$, $(NH_4)_2SO_4$ and $NH_4HSO_4$, were highly water-soluble (Pakkanen et al., 1996; Foltescu et al., 1996;

Seinfeld and Pandis, 1998; Moise et al., 2002) and these bio-available inorganic salts decomposed immediately and rapidly into ions forms in contact with liquids (Lide 2007). However, considered that averagely 33 % (15-53%) of the cumulative dissolved $PO_4^{3-}$ and 34% (5%-69%) of the cumulative dissolved $SiO_3^{2-}$ in aerosols were dissolved at pH 7.8, the dissolution of aerosol P and Si species were



induced by both nutrient concentration differences and chemical reactions that caused by ion attractions,

which meant more complex dissolution patterns than N species.

Typically 90% water-soluble components of aerosols over the Yellow sea were $NH_4^+$, $NO_3^-$, $SO_4^{2-}$ and $Cl^-$ (Yang & Xiu, 2009; Wang et al., 2018). Assumed that $NH_4Cl$, $NH_4NO_3$, $(NH_4)_2SO_4$ were major sources of $NH_4^+$ and $NO_3^-$ ions and dissolved in stable stoichiometric relation, $NH_4^+/NO_3^-$ ratio must be larger than 1. If considered the aging process in the transport pathway of aerosol particles, there existed

a tendency to form complex salt $((NH_4NO_3)_3 \cdot (NH_4)_2SO_4$ and $(NH_4NO_3)_2 \cdot (NH_4)_2SO_4)$ (Schlenker et al., 2003; Saathoff et al., 2003) except for gas-particle transformation forms $((NH_4)_2SO_4$ and $NH_4NO_3)$. In a more complex situation, $NH_4^+/NO_3^-$ ratios were also larger than 1 under assumption that $NH_4Cl$, $NH_4NO_3$, $(NH_4)_2SO_4$, $(NH_4NO_3)_3 \cdot (NH_4)_2SO_4$ and $(NH_4NO_3)_2 \cdot (NH_4)_2SO_4$ were primarily providers of $NH_4^+$ and $NO_3^-$. However, cations such as $Ca^{2+}$ and $Na^+$ were also composed of aerosols (Suzuki et al.,

2010; Zheng et al., 2011). Since spr-NW aerosol had highest Ca-P/TP (55%) and win-NW2 aerosol had lowest Ca-P/TP (20%), these two aerosols were selected for testing whether $NH_4^+/NO_3^-$ ratio could be an indicator to aerosol major components under the assumption that Ca were mostly combined with P since apatite minerals were precursors in P aerosols. Accumulated $NH_4^+/NO_3^-$ ratios (Fig. 8) for leachates from start till $NH_4^+/NO_3^-$ ratio curve reached the plateau were 0.9±0.1 and 2.6±0.3 on average

for spr-NW and win-NW2 aerosols in aged seawater and 0.4±0.01 and 1.5±0.1 for spr-NW and win-NW2 aerosols in Milli-Q water. Accumulated $NH_4^+/NO_3^-$ ratio curves may indicated that main components of win-NW2 sample were $NH_4Cl$, $NH_4NO_3$, $(NH_4)_2SO_4$ and/or their complex salts, while contents of other inorganic salts containing contained $NO_3^-$ in the spr-NW aerosol such as $Ca(NO_3)_2$ and $NaNO_3$ that produced from the gas-particle conversions were not negligible, which was in consist with

reported result that spr-NW aerosol carried terrigenous particles and/or loess soils (Liu et al., 2002). In summary, $NH_4^+/NO_3^-$ ratio could be a rough indicator to evaluate crustal element such as Ca as major components.

Aerosol P and Si species were mainly composed of soil-derived crystalline minerals, artificial




fertilizer particles and their aged amorphous forms (Yuan et al., 2018) because they can not be formed

by gas-particle transformation. In addition, physical and/or chemical changes of aerosols could occur

via atmospheric transport paths. Therefore, superficial exchangeable P and Si species (aged amorphous

forms) that were primarily affected by surface diffusion rates may be the first components released into

solution (Tang et al., 2003; Wang et al., 2017). After superficial exchangeable P and Si had been used

up within 45min, crystalline particles started to collapse resulting from $H^+$ ion attack effect in exchange

for P and Si from solids (Guidry and Mackenzie, 2003). At pH 5.5, $H^+$ concentrations were not strong

enough to easily drag P and Si out of their lattice. Besides, leaching system was entirely not acid- or

alkline-buffered, so if solid $CaCO_3$ had not been consumed up, $H^+$ would first attacked $CaCO_3$ before

attacked P-containing and Si-containing particles, resulting in the lesser release of P and Si into solution.

As shown in dissolution equations, $k_2$ of P and Si in aerosol were small constants, which supported that

exchangeable P and Si had already dissolved; and n was 0, which demonstrated that dissolution

reactions were independence from reactant concentrations. At pH 2.0, $H^+$ concentrations were sufficient

to break up crystalline structures of aerosol particles and liberate them into liquid phase and no more P

and Si release in aerosols would occur once soil-derived crystalline minerals and anthropogenic

fertilizers were depleted by $H^+$ in leaching liquids.

Average k values of P and Si dissolution for Qianliyan aerosols were 0.25±0.08 and 0.22±0.06,

respectively, and average k values of P and Si dissolution for Qingdao aerosols were 0.10±0.11 and

0.16±0.07, respectively. Comparison showed that dissolution rate constants were close, however,

dissolution rates for Qianliyan and Qingdao aerosols were varied. Also, water-soluble/acid-soluble P

and Si ratios in Qingdao aerosols showed apparent discrepancy (Wang et al., 2017), but they showed a

rather constant values in Qianliyan aerosols (Fig. 4). When considering that dissolution reactions of P

and Si were quasi-first-order and the slowest reactions controlled reaction rates, the least soluble

P-containing and Si-containing components in Qianliyan and Qingdao aerosols were different.

Alternations of aerosol P-containing and Si-containing precursors surface properties during atmospheric



mass transport, such as hygroscopicity and crystal lattice deformation (rough surface) that could

possibly make the size changes of P-containing and Si-containing particles were resulted in variations

that water-soluble P and Si needed around 40 min to reach dissolution equilibrium for Qianliyan

aerosols, but around 15 min for Qingdao aerosols.

  Natural apatites and silicates minerals were major precursors of aerosol particles. Generally, apatites

included $Ca_5(PO_4)_3F$, $Ca_5(PO_4)_3OH$, $Ca_5(PO_4)_3Cl$ and silicates included quartz, montmorillonite, illite

and kaolinite etc. Laboratory dissolution rate experiments on minerals indicated that silicates dissolution

rates were often lower than that of apatites (Lerman & Wu, 2008). Mineral dissolution rates were often

normalized by total surface area, so mean dissolution rate ($r_a$) was defined as $r_{ai} = \dfrac{c_e}{c_{TSP} * s * t_e}$ , where i

was the stage 1, 2 and 3 corresponding to scenarios at pH 7.8, 5.5 and 2.0, respectively, s was aerosol

surface area, $c_e$ was the accumulated concentrations when dissolution reached equation, $c_{TSP}$ was aerosol

TSP mass concentration and $t_e$ was time when dissolution reached equation. It assumed that aerosol

surface area (s) was $4 m^2 \cdot g^{-1}$ (Guo et al., 2017) and $r_{ai}$ could compare with these mineral dissolution rates

of aerosol precursors after normalization. SW aerosol had higher $r_a$ than other NW aerosols (Table S4)

indicated that mineralogical compositions were source-induced different. The largest $r_{a1}$ of P-containing

aerosols was $1562 \cdot 10^{-12}$ $mol m^{-2} s^{-1}$, which was much lower than that of carbonated hydroxyapatite

($52000 \cdot 10^{-12}$ $mol \cdot m^{-2} s^{-1}$; Tang, et al., 2003) and was close to that of igneous fluorapatite ($1830 \cdot 10^{-12}$

$mol \cdot m^{-2} s^{-1}$; Valsami-Jones, et al., 1998); the lowest $r_{a1}$ was $60 \cdot 10^{-12}$ $mol \cdot m^{-2} s^{-1}$, which was much higher

than that of sedimentary carbonate fluorapatite ($5.01 \cdot 10^{-12}$ $mol \cdot m^{-2} s^{-1}$; Guidry & Mackenzie, 2003) and

was close to that of pure hydroxyapatite ($53.3 \cdot 10^{-12}$ $mol \cdot m^{-2} s^{-1}$; Valsami-Jones, et al., 1998); the mean $r_{a1}$

was $443 \cdot 10^{-12}$ $mol \cdot m^{-2} s^{-1}$, which was slight higher than that of natural fluorapatite ($380 \cdot 10^{-12}$ $mol \cdot m^{-2} s^{-1}$;

Guidry & Mackenzie, 2003). Aerosol mean $r_{a1}$ was close to dissolution rate of fluorapatite mineral,

indicating that fluorapatite was possibly main inorganic composition in all aerosols. Aerosol mean $r_{a1}$

for Si were much higher than mineral dissolution rates of most silicates, while the largest $r_{a1}$ of



Si-containing aerosol particles was $3916·10^{-12}$ $mol·m^{-2}s^{-1}$, the lowest $r_{a1}$ was $23·10^{-12}$ $mol·m^{-2}s^{-1}$ and the mean $r_{a1}$ was $838·10^{-12}$ $mol·m^{-2}s^{-1}$. Also, summer and winter aerosols had lower $r_a$ than other seasons

aerosols. Laboratory dissolution rates of pure quartz was $15·10^{-12}$ $mol·m^{-2}s^{-1}$, major mineral clay silicates, such as kaolinite, montmorillonite and illite, were $0.012·10^{-12}$, $9.8·10^{-12}$ and $0.011·10^{-12}$ $mol·m^{-2}s^{-1}$, respectively (Huertas et al., 1999; Bauer and Berger ,1998; Köhler et al., 2003). These low dissolution rate species could not support higher aerosol $r_{a1}$ since mixed dissolution rates were controlled by the lowest reaction reactions. Dissolution rates of Ca- and Mg-containing silicates, such as

enstatite, diopside, olivine ranged from $10\text{-}316.2·10^{-12}$ $mol·m^{-2}s^{-1}$ (Oelkerset al., 2001; Schott et al., 1981; Lasaga, 1988). These slight higher dissolution rate species could be possibly considered as majority Si-containing aerosol species in summer and winter. However, the highest dissolution rate species, pure $CaAl_2Si_2O_8$ and anorthite were $10000·10^{-12}$ and $2820·10^{-12}$ $mol·m^{-2}s^{-1}$ (Fleer, 1982; Blum and Stillings, 1995), of which anorthite could be considered as majority Si-containing aerosol species in

spring and autumn. Despite of different mineralogical compositions, dissolution rate discrepancies between aerosols and their natural precursors may be explained by atmospheric radial or acidified aging process (Mashburn, 2006; Cuadros et al., 2015). Natural minerals acted as an important reactive surface that providing a heterogeneous sink for acidic gas species, dramatically changed the morphology of airborne particles, resulting in expected greater dissolution rates of aerosols. Given that carbonates

dissolution rates were incredibly 4-6 orders of magnitude higher than dissolution rates of silicates, carbonated hydroxyapatite dissolution rate was 1-4 orders of magnitude higher than other apatites, hence, co-dissolution of carbonated species might accelerate the rate of aerosol particles dissolution rates and accumulated dissolution amount.

Refractory element Fe was reported to be synchronously released with easier and earlier dissolution

of nitrate salts and other acids (sulphate salts and organic acids) and release of Fe and nitrate salts in Milli-Q were proportioned because of their tight morphology combination or same source (Hsu et al., 2013), but Ca was preferentially released compared to P from the mineral structure ($Ca_5(PO_4)_3F$ and





Ca$_5$(PO$_4$)$_3$OH) at early dissolution reaction (Guidry and Mackenzie, 2003). Theoretically, Fe-P, Ca-P and De-P all could participate in dissolution process, except for OP. Hence, challenges that

distinguishing the typical dissolution parameters of individual P mineral species were failed to overcome even specify P morphology forms unless increasing sample numbers or changing the collection mode into single particle aerosol.

### 4.2. Control Factors of Phosphorus Dissolution

Aerosol source, solution pH and solution ion strength (IS) were three major factors that controlled P

solubility in certain leaching liquids. P solubility were also varied with solvents though it had not been discussed afterward.

Firstly, factor affected P solubility may ascribed to source. Regardless of the temporal and spatial influences, P solubility in SW aerosols were generally larger than NW aerosols, for example, SW aerosol had highest aged seawater-soluble P solubility (39%) than NW aerosols (16% ± 11%) at

Qianliyan and SW aerosols had higher Milli-Q water-soluble P solubility (33% ± 10%) than NW aerosols (16% ± 7%) at Qingdao as well (Wang et. al., 2017). P solubility of source soils may be different in itself for different ability on fixing water-soluble phosphorus. Generally, NW-source soils (such as black soil and loess) were weakly alkaline and SW-source soils (such as red soil) were acidic. If acid had the same mechanism for soil P solubility and aerosol P solubility, P solubility in SW aerosols

were surely larger than NW aerosols. However, source-induced variation also inherently included aging processes during airborne mass transport, but the degree of aerosol aging was difficult to quantify. Therefore, future research needs to further divest the intrinsic core of source factor.

Secondly, factor affected P solubility may attributed to acidity. Milli-Q acid-soluble P solubility was three times higher than Milli-Q water-soluble P solubility and aged seawater acid-soluble P solubility

was approximately three times higher than aged seawater-soluble P solubility on average (Fig. 7), which highlighted the acidity effect. Laboratory simulated atmospheric acidification treatment caused a 10-40





times increase of soluble phosphorus amount (Nenes et al., 2011) in Sahara soil dust, however, in our experiment, only 5 times increase of soluble phosphorus amount for spr-NW aerosol that strongly influenced by Asian sand-dust event. This showed variations of acidity effect on different regional

aerosols. Spr-NW aerosol that had been acidified in the leaching experiment displayed no effect under slight increase of $H^+$ ($10^{5.5}$ mol·$L^{-1}$), but a sharp and solid increase amount of soluble phosphorus under stronger $H^+$ ($10^{2.0}$ mol·$L^{-1}$). This implied that particulate P in aerosol accelerated dissolution when leaching solution reached a given $H^+$ concentration. Study on atmospheric acid processing of mineral dust brought up the point that less than 10% of total phosphorus dissolved when $H^+$ ion concentration

was under $10^{-4}$ mol $H^+$/g of dust but when $H^+$ ion concentration was higher than $10^{-4}$ mol $H^+$/g of dust, aerosol particulates consumed $H^+$ ions until inorganic form of phosphorus particles exhausted (Stockdale et al., 2016). In our high time-resolution dissolution experiments, hypothesized that the mass of aerosol on subsamples unchanged during the whole processing because only the soluble species removed, $H^+$ ion concentration for spr-NW aerosol was $1.8 \cdot 10^{-7}$ mol $H^+$/g at pH 7.8 and $2.4 \cdot 10^{-5}$ mol

$H^+$/g at pH 5.5. These $H^+$ ion concentrations were below the threshold value. But $H^+$ ion concentration that was $9.6 \cdot 10^{-2}$ mol $H^+$/g at pH 2.0 was exceeded the threshold value. Though particle properties were different between Sahara dust and Asian dust, threshold of $H^+$ ion per particle did existed and further dissolution did occur only when acidity surpassed the threshold value. In future, whether Sahara dust and Asian dust shared the same threshold or different threshold should be tested.

Thirdly, factor affected P solubility may be attributed to a small extent to IS. The effect of seawater or salt addition on P solubility were positive because aged seawater acid-soluble P solubility was averagely slightly higher than Milli-Q acid-soluble P solubility. In theory, presence of strong electrolytes (NaCl and $MgCl_2$) in aged sea water strengthened electrostatic interactions (such as desorption) and liberated rather insoluble masses into solution. However, IS-induced effect was smaller than source and pH

effects in high time-resolution dissolution experiment because aged seawater (IS=0.7) increased P solubility by a factor of 1-2, unless adding NaCl to increase IS to 2, it was possible to increase P





solubility by a factor of 4.3 (Stockdale et al., 2016).

## 5. Conclusions

In this study, we mainly focused on the high time-resolution dissolution experiments, modeled the
dissolution curves and probed factors influenced dissolution rates, especially for P solubility. Our
modeling of nutrient dissolution curves depicted by Avrami model and/or zero-order kinetics model for
high time-resolution dissolution experiments were indicated that reactions of aerosol inorganic N, P and
Si species short-time dissolution in natural waters were quasi-first-order. Varied dissolution rates were
caused by aerosol inorganic components. Discussions of aerosol inorganic components and dissolution
patterns revealed their inherent relations that N species were more likely to immediately and completely
dissolve due to fine particles formed by gas-particle transformation, while P and Si were tended to
dissolve more in low acidity largely because of coarse soil-derived mineral particles. By comparisons of
water-soluble and acid soluble nutrient concentrations in dissolution experiments and ultra-sound
extractions, ratios of acid-soluble to water-soluble nutrient concentrations were 1.0 (0.9-1.1) for $NH_4^+$
and $NO_3^-$, 2.4 (2.1-2.6) for $PO_4^{3-}$ and 2.5 for $SiO_3^{2-}$. In addition, prominent factors influenced P
solubility were source and acidity, while other less important factors that could affected aerosol P
solubility were solution ion strength.

### Acknowledgements

This paper is financially supported by the National Research and Development program of
China (2016YFA0600902), Aoshan Talents Program Supported by Qingdao National Laboratory for
Marine Science and Technology (2015ASTP-OS08), National Science Foundation of China (41521064,
41776087), and the Taishan Scholars Program of Shandong Province. We also sincerely thank the staffs
working in the Qianliyan Department of the North China Sea Branch of State Oceanic Administration
for their help in sample collection.

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





# Appendices

**Figures:**

**Fig. 1.** -72h back-trajectories of air-masses of six aerosols for high time-resolution nutrient dissolution experiments.

**Fig. 2.** A schematic diagram of high time-resolution dissolution experiment.

**Fig. 3.** (a) Concentrations of P species in aerosols by SEDEX method. (b) Relative fractions of inorganic P species.

**Fig. 4.** Linear relationships between water-soluble and acid-soluble (a) $NO_3^-$ and $NH_4^+$ concentrations and (b) $PO_4^{3-}$ and $SiO_3^{2-}$ concentrations in ultra-sound extractions (square) and high time-resolution dissolution experiments (triangle).

**Fig. 5.** Accumulated dissolution curves for high time-resolution dissolution experiments.

**Fig. 6.** Accumulated P/Si ratio curves for high time-resolution dissolution experiments.

**Fig. 7.** P solubility curves for high time-resolution dissolution experiments.

**Fig. 8.** Accumulated $NH_4^+/NO_3^-$ ratio curves for high time-resolution dissolution experiments.





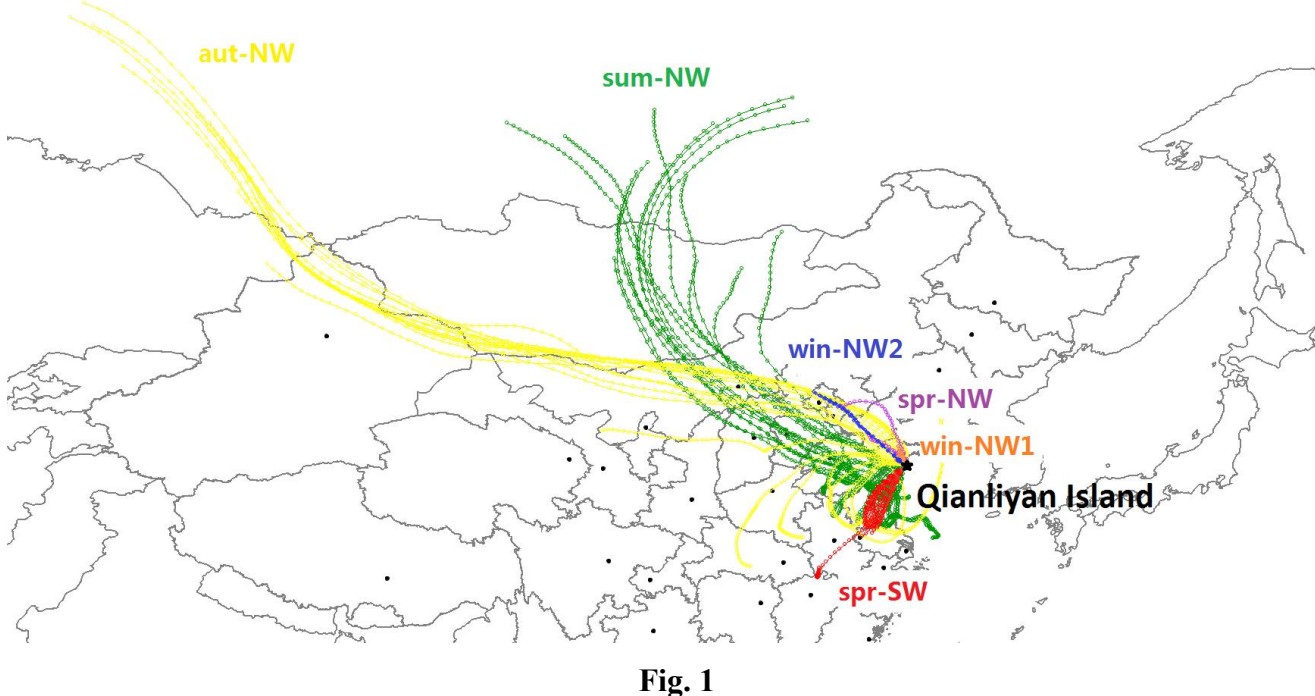

**Fig. 1**





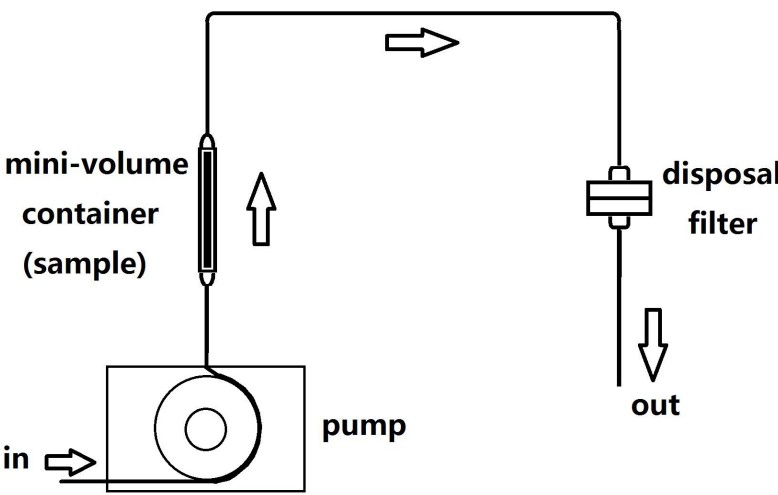

**Fig. 2**




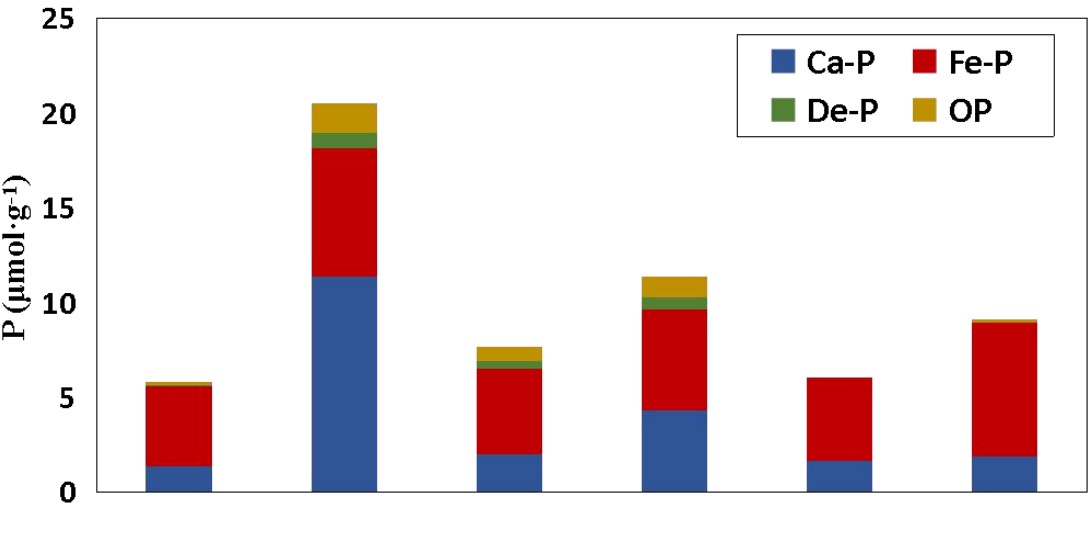

(a)

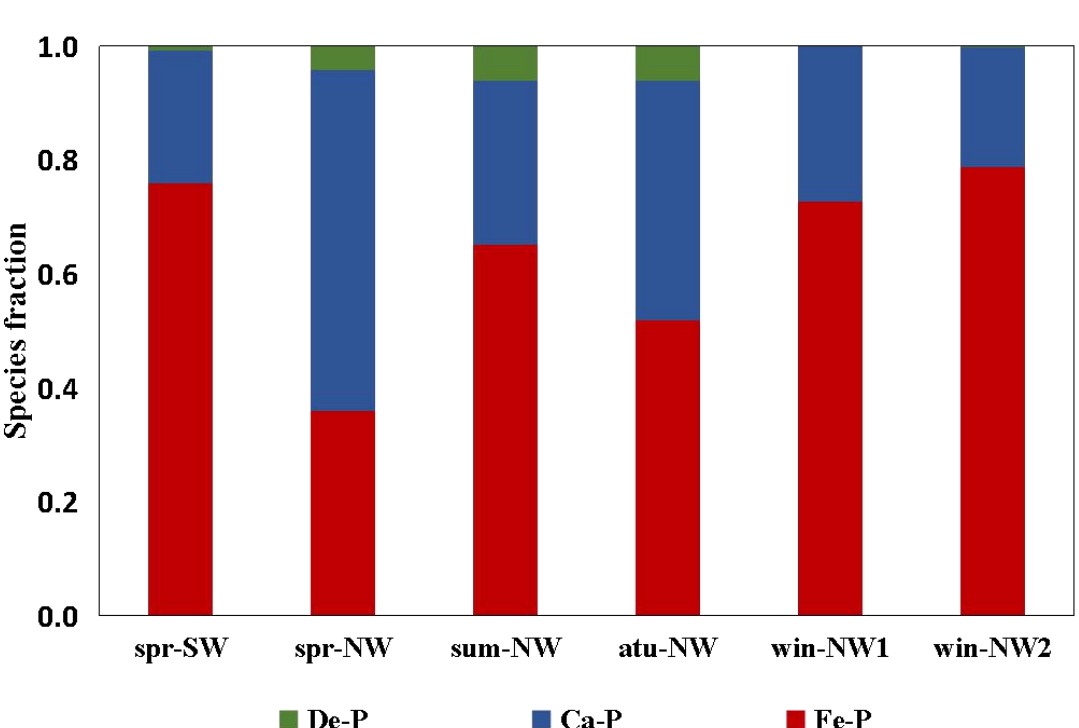

(b)

**Fig. 3**




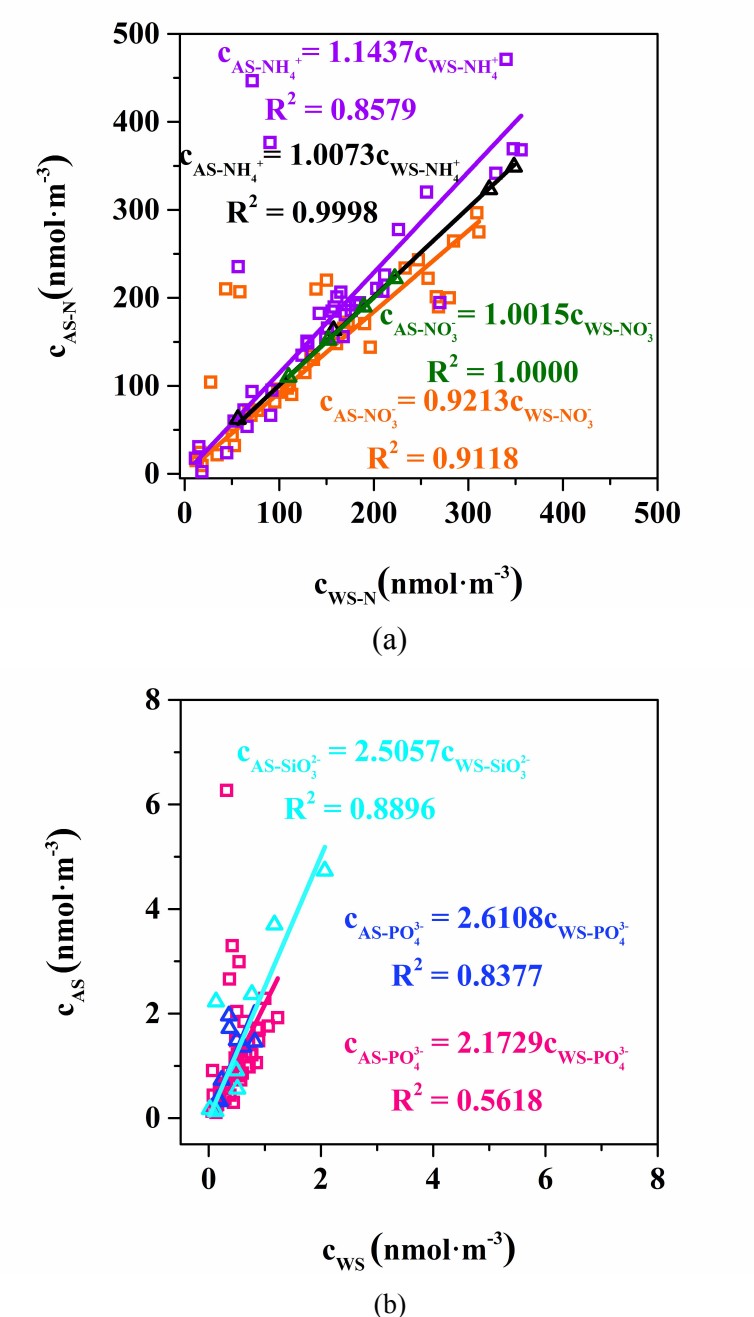

(a)

(b)

Fig. 4






**Fig. 5**

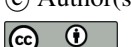



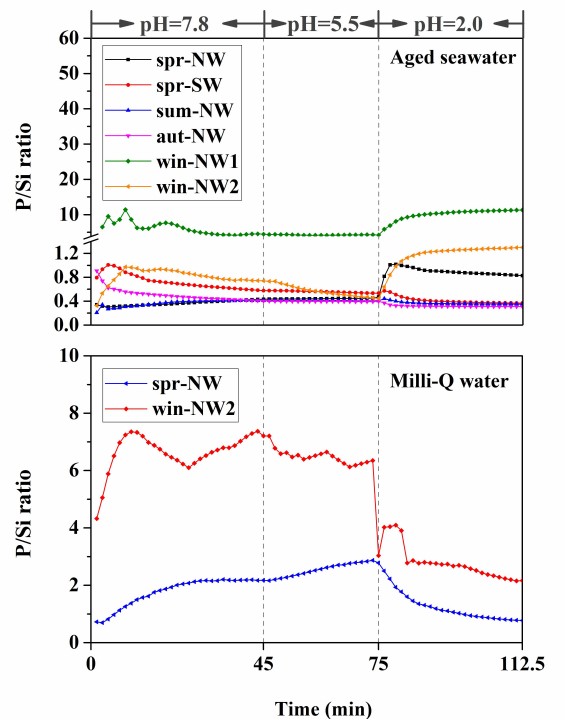


**Fig. 6**

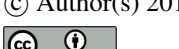



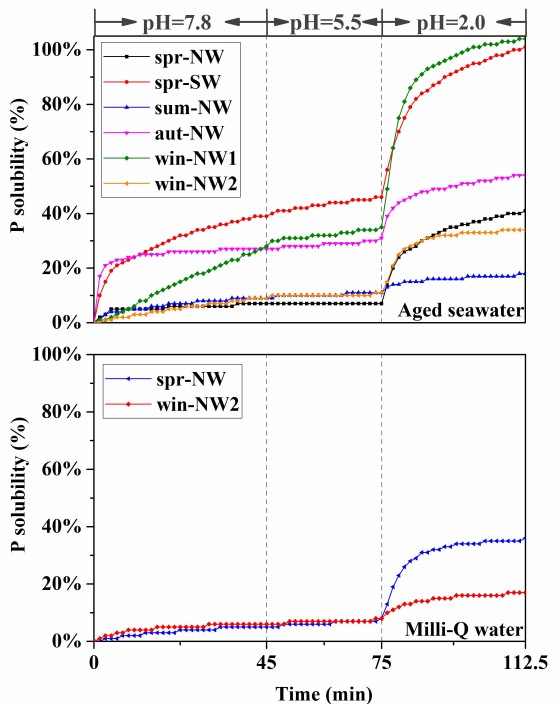

**Fig. 7**




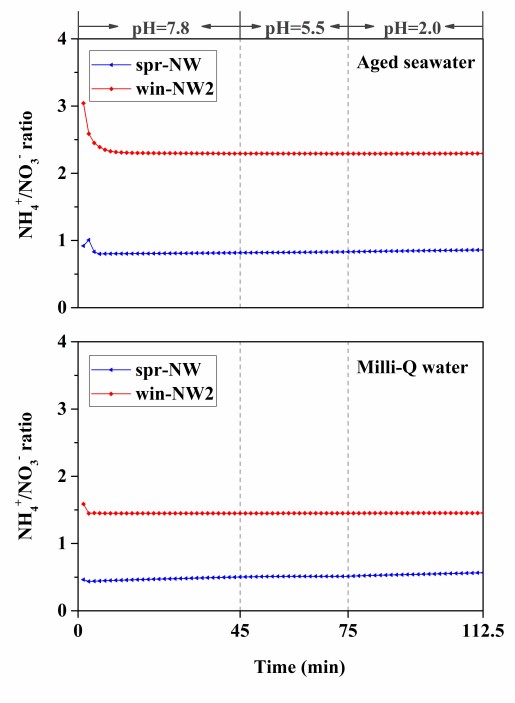

**Fig. 8**



**Table 1.** One-stage aerosol $NH_4^+$ and $NO_3^-$ dissolution kinetic parameters for high time-resolution dissolution experiments, which included dissolution rate constant (k, $min^{-1}$), reached dissolution equilibrium time ($t_e$, min) and accumulated $NH_4^+$ and $NO_3^-$ concentrations ($nmol·m^{-3}$) at dissolution equilibrium time ($c_e$), pH 7.8 ($c_{45}$), pH 5.5 ($c_{75}$) and pH 2.0 ($c_{112.5}$).

| | k | n | $t_e$ | $c_e$ | $c_{45}$ | $c_{75}$ | $c_{112.5}$ |
|---|---|---|---|---|---|---|---|
| Milli-Q water-soluble $NH_4^+$ | | | | | | | |
| spr-NW | 1.24 | 0.32 | 6 | 48.5 | 55.1 | 56.3 | 62.3 |
| win-NW2 | 1.65 | 0.56 | 6 | 318.6 | 321.8 | 322.1 | 323.2 |
| Aged seawater-soluble $NH_4^+$ | | | | | | | |
| spr-NW | 0.44 | 1.31 | 6 | 151.0 | 154.7 | 157.4 | 163.1 |
| win-NW2 | 0.64 | 0.68 | 6 | 307.0 | 347.8 | 348.3 | 349.1 |
| Milli-Q water-soluble $NO_3^-$ | | | | | | | |
| spr-NW | 1.31 | 0.79 | 7.5 | 109.5 | 109.8 | 109.9 | 110.1 |
| win-NW2 | 1.27 | 0.71 | 7.5 | 220.8 | 222.1 | 222.1 | 222.3 |
| Aged seawater-soluble $NO_3^-$ | | | | | | | |
| spr-NW | 0.25 | 1.67 | 7.5 | 189.1 | 189.3 | 189.4 | 189.9 |
| win-NW2 | 0.40 | 0.88 | 7.5 | 136.5 | 151.6 | 152.0 | 152.2 |






**Table 2.** Three-stage aerosol $PO_4^{3-}$ and $SiO_3^{2-}$ dissolution kinetic parameters for high time-resolution dissolution experiments, which included dissolution rate constant ($k_1$, $k_2$ and $k_3$, $min^{-1}$), reached dissolution equilibrium time ($t_{e1}$ and $t_{e3}$, min) and accumulated $PO_4^{3-}$ and $SiO_3^{2-}$ concentrations ($nmol \cdot m^{-3}$) at dissolution equilibrium time ($c_{e1}$ and $c_{e3}$), pH 7.8 ($c_{45}$) , pH 5.5 ($c_{75}$) and pH 2.0 ($c_{112.5}$). "-" represented a special case that win-NW1 sample had no further $SiO_3^{2-}$ dissolution in strong acidic aged seawater.

| | $k_1$ | $n_1$ | $t_{e1}$ | $c_{e1}$ | $c_{45}$ | $k_2$ | $c_{75}$ | $k_3$ | $n_3$ | $t_e$ | $c_{e3}$ |
|---|---|---|---|---|---|---|---|---|---|---|---|
| Milli-Q water-soluble $PO_4^{3-}$ | | | | | | | | | | | |
| spr-NW | 0.05 | 1.01 | 37.5 | 0.23 | 0.26 | 0.054 | 0.26 | 0.20 | 0.88 | 106.5 | 1.69 |
| win-NW2 | 0.14 | 0.78 | 37.5 | 0.13 | 0.14 | 0.008 | 0.16 | 0.21 | 0.76 | 106.5 | 0.35 |
| Aged seawater-soluble $PO_4^{3-}$ | | | | | | | | | | | |
| spr-SW | 0.22 | 0.66 | 37.5 | 0.50 | 0.53 | 0.030 | 0.62 | 0.15 | 0.85 | 106.5 | 1.33 |
| spr-NW | 0.37 | 0.53 | 37.5 | 0.32 | 0.33 | 0.064 | 0.36 | 0.15 | 0.85 | 106.5 | 1.90 |
| sum-NW | 0.13 | 0.78 | 37.5 | 0.16 | 0.17 | 0.005 | 0.20 | 0.27 | 0.62 | 106.5 | 0.31 |
| atu-NW | 0.90 | 0.42 | 37.5 | 0.72 | 0.73 | 0.025 | 0.82 | 0.34 | 0.59 | 106.5 | 1.43 |
| win-NW1 | 0.01 | 1.44 | 42 | 0.37 | 0.40 | 0.040 | 0.50 | 0.34 | 0.59 | 106.5 | 1.47 |
| win-NW2 | 0.02 | 1.19 | 42 | 0.19 | 0.20 | 0.020 | 0.23 | 0.22 | 0.85 | 106.5 | 0.72 |
| Milli-Q water-soluble $SiO_3^{2-}$ | | | | | | | | | | | |
| spr-NW | 0.32 | 0.49 | 39 | 0.11 | 0.12 | 0.084 | 0.13 | 0.04 | 1.16 | 106.5 | 2.05 |
| win-NW2 | 0.32 | 0.54 | 22.5 | 0.02 | 0.02 | 0.005 | 0.05 | 0.10 | 0.83 | 106.5 | 0.15 |
| Aged seawater-soluble $SiO_3^{2-}$ | | | | | | | | | | | |
| spr-SW | 0.10 | 0.77 | 37.5 | 0.82 | 0.92 | 0.101 | 1.17 | 0.05 | 1.19 | 106.5 | 3.57 |
| spr-NW | 0.20 | 1.87 | 7.5 | 0.76 | 0.77 | 0.064 | 0.77 | 0.03 | 1.26 | 106.5 | 2.24 |
| sum-NW | 0.33 | 0.54 | 37.5 | 0.38 | 0.41 | 0.017 | 0.49 | 0.10 | 0.95 | 106.5 | 0.89 |
| atu-NW | 0.28 | 0.59 | 37.5 | 1.69 | 1.80 | 0.106 | 2.07 | 0.30 | 0.65 | 106.5 | 4.63 |
| win-NW1 | 0.00 | 1.52 | 37.5 | 0.08 | 0.09 | 0.001 | 0.12 | - | - | - | - |
| win-NW2 | 0.03 | 1.01 | 42 | 0.25 | 0.27 | 0.002 | 0.51 | 0.13 | 0.96 | 106.5 | 0.56 |




**Table 3.** Person correlation coefficients for aged seawater-soluble P solubility (SWPS, %), aged
seawater acid-soluble P solubility (SAPS, %), dissolution rate constant ($k_1$, $k_2$ and $k_3$, $min^{-1}$), TSP mass
concentration ($\mu g \cdot m^{-3}$) and TP, Fe-P, Ca-P and De-P concentrations ($nmol \cdot m^{-3}$).

|  | SWPS | SAPS | k1 | k2 | k3 | TSP | TP | Fe-P | Ca-P | De-P |
|---|---|---|---|---|---|---|---|---|---|---|
| **SWPS** | 1 | | | | | | | | | |
| **SAPS** | 0.872* | 1 | | | | | | | | |
| **k1** | 0.197 | -0.112 | 1 | | | | | | | |
| **k2** | 0.098 | 0.387 | 0.212 | 1 | | | | | | |
| **k3** | 0.149 | 0.072 | 0.229 | -0.278 | 1 | | | | | |
| **TSP** | -0.755 | -0.547 | 0.088 | 0.532 | -0.448 | 1 | | | | |
| **TP** | -0.556 | -0.439 | 0.403 | 0.642 | -0.364 | 0.934** | 1 | | | |
| **Fe-P** | -0.677 | -0.539 | 0.066 | 0.35 | -0.357 | 0.829* | 0.705 | 1 | | |
| **Ca-P** | -0.459 | -0.314 | 0.385 | 0.719 | -0.379 | 0.893* | 0.984** | 0.588 | 1 | |
| **De-P** | -0.434 | -0.541 | 0.688 | 0.317 | -0.071 | 0.654 | 0.834* | 0.348 | 0.825* | 1 |

* Significant at 0.01

** Significant at 0.01