# Peer review of "Nutrients Dissolution Kinetics of Aerosols at Qianliyan Island, the Yellow Sea by a High Time-resolution Nutrient Dissolution Experiment, Potential Linkages with Inorganic Compositions and P solubility controlled factors"

_Atmospheric Chemistry and Physics, 2018_

## Referee Comment (RC1) · Anonymous Referee #2 · 20 Dec 2018

Review of Zhang et al This paper describes an extensive series of experiments on the solubility of N, P and Si from some aerosols collected off the coast of China. The work has been carefully done and thoroughly analysed and should be published. The paper is generally well written but would benefit from an additional edit in a few places. The discussion does seem to me to be too long in some places. The authors discuss the detailed comparisons of a few aerosols from this one site, but most readers will be more interested in the generalities of the results which I think can be summarised

as inorganic N dissolves completely and rapidly, while only a percentage of P and Si dissolves and that dissolution takes place over timescales of a few hours. I would also suggest that the authors need to note a few caveats of this type of experiment. Firstly wet deposition dominates in most places. The pH of aerosol or rain depositing to sea-water will rise to close to 8 almost immediately so prolonged acid exposure can happen in clouds but will then rapidly be reversed. The timescales of dissolution relevant to marine ecosystems are the lifetimes in the surface mixed layer of particles which are days to weeks so the dissolution rates of even the Si and P specie rea rapid with respect to that. A few specific comments Line 100 Explain why weekly collections only span 20 hours, I assume it is collecting for 1 day each week. Line 120 why was 1M HCl used, that is surely very much more acidic. In addition the P and Si analyses methods are sensitive to the pH of the analysed solution, did this cause any issues? Line 226 Why do you link the FeP pattern to acid processing rather than source, it seems to me it could be either. Line 280-5 and 328-358 These are quite long discussion sections that could be shortened to focus on key and generalizable conclusions rather than specific comparisons of a few aerosols from this one site – consider the wider implications for readers form outside the region. The issue of comparisons of rates of dissolution for pure minerals, particularly silicates, to the observed rates (around line 380) are an interesting observation that should be retained. Line 307 The logic of the discussion around nitrate/ammonium ratios seems to me to be a bit flawed. At the very least it ignores sodium nitrate formed by the seasalt displacement reaction in the coarse mode aerosol. The nitrate/ammonium ratio in an aerosol depends on emission rates and deposition so I'm not sure this section is particularly useful.

---

## Short Comment (SC1) · 30 Dec 2018

After reading this manuscript, I find that there are serious issues in this study. Qianliyan Island was only one very small sampling site. In more marked deficiencies, only 6 aerosol samples were collected during a short period at this small island. Research area is most important for atmospheric science, but I have no idea from this manuscript. Atmospheric Chemistry and Physics (ACP) should publish a large scale study with a

wide range of sampling fields (e.g., covering the whole Yellow Sea and its coastal areas) and enough samples (e.g., at least 12 months samples which covered a whole year). Therefore, this limited research is not suitable to be published in ACP.

These 6 aerosol samples capturing seasonal dominant aerosol sources were selected to carry out the high time-resolution dissolution experiments. It was not clear why only six samples were selected. The 6 aerosol samples are indeed too few to study temporal variation. Moreover, the significance of this research site was not described in detail. Sampling information (including sampling location, date, time etc.) of six samples are vital to data analyze and discussion; unfortunately, they were not described.

All data necessary to understand, evaluate, replicate, and build upon the reported research must be made available and accessible whenever possible. Critically, I do not see any reference to the air pressure, temperature, wind direction, wind speed, relative humidity, TSP, nutrients concentrations data, not even in the table.

Lines 615-623: Three references cites as Chen et al. 2006 were shown. If two or more references from the same year contain the same first six or more authors, use a, b, c, and so on for the in-text citation and in the references list. Please check ACP reference format.

It is noted that this manuscript needs careful editing by someone with expertise in technical English editing paying particular attention to English grammar, spelling, and sentence structure so that the goals and results of the study are clear to the reader.

---

## Short Comment (SC2) · 30 Dec 2018

In this study, a series of nutrient dissolution experiments were conducted to determine the soluble fraction of atmospheric nutrients and revealed the short-time dissolution processes, patterns and kinetics of nutrient elements in aerosols. However, I found that the method used to collect aerosol samples had a significant drawback. Total suspended particulate (TSP) samples were collected using poly-carbonate filters in 2011

and Whatman cellulose fiber filters in 2012. Why were different filters used in collecting TSP samples in 2011 and 2012? Was the influence of different filters assessed before the use? This is an important issue because it will largely impact nutrients dissolution kinetics of aerosols and their controlling factors. Supposing both filters are suitable for TSP collection and chemical analysis, but the authors should have used the same kind of filter in the same sampling site for the collection of different season samples. This is essential to maintain the reliability and comparability of the data. Moreover, the sample number (6) in this study is too small to interpret temporal change of aerosol dissolution processes. More than that, I did not find the detailed processes and mechanism on nutrients elements dissolution in aerosols in the present study. The authors should carefully address this issue in more details. Overall, this study is a local investigation in a small island but not focused on studies with general implications for atmospheric science. I do not feel that this manuscript fits the scope of Atmospheric Chemistry and Physics due to its too local interest and its defect in sample collection methods.

---

## Referee Comment (RC2) · Anonymous Referee #1 · 2 Jan 2019

Zhang et al. assessed dissolution of three nutrients (N, P, Si) from six aerosol samples at Qianliyan Island, the Yellow Sea. The results could have useful implications for nutrient availability to marine ecosystems from dry deposition. However, the methodology of this work has several drawbacks, which needs to be addressed properly. First, the aerosol samples were collected from only one island site and there were no replicates for sampling sites. The authors Second, total suspended particulate (TSP) samples were collected by using two different filters in 2011 and 2012 (the poly-carbonate fil-

ters in 2011 and Whatman cellulose fiber filters in 2012). I am wondering if these different filters could affect the particle size and composition of the aerosol samples. This should be addressed by an experimental approach. Third, the authors collected 39 aerosol samples but they only measured nutrient dissolution from six aerosol samples. I understand that the authors intended to analyze samples by season and main source. However, it is a pity that the nutrient dissolution experiment has no replicates for each category of samples (spr-SW, spr-NW, sum-NW, aut-NW, win-NW1 and win-NW2). This hinders comparison between sample categories by a statistical analysis. Moreover, the manuscript also requires extensive editing for English language including grammar and word choice. Overall, I don't think this manuscript should be considered for publication at this moment.

Specific comments

L1: Title: The current title is not appropriate for an ACP submission. It needs to be shortened. L26-28: It is likely a speculation that is not well supported by current results. L101-118(Figure 1): Please give more information on the calculation of backward trajectories, including input data, methodology and validation. Figures, 3, 5-8: No replicates and no statistical comparison between aerosol sample categories.

---

## Short Comment (SC3) · 27 Jan 2019

Thank you for your comment. First, so far, only the comparison between fiber filter and quartz filter on nutrient sampling and analysis (Pszenny et al., 1993) has been reported and there are no related reports about the comparison of the polycarbonate filters and cellulose fiber filters, it acquiesced that there is no difference between them on aerosol bulk sampling for nutrients analysis. None sole use of filter was an unavoidable pity

because of the shortage of polycarbonate film in the market and recommendation of cellulose fiber film in national standard. Second, the sample size was small, however, they basically captured the seasonal characteristics of the main source direction. And our high resolution experiments showed on the aerosol kinetics dissolution process, the maximum dissolution amount and the proof of the changes in aerosol during atmospheric processes by dissolution rate deviations from phosphorus and silicon minerals. I agreed that the study on more detailed processes and mechanism should be worked out in the near future with the help of sing particle collection, high-resolution electron microscope and other techniques. Third, although the geographical location affects the dissolution rate of aerosol, the general pattern, such as first-order dissolution reaction of short-time dynamic dissolution of aerosols in seawater, will not change despite of these limitations.

---

## Short Comment (SC4) · 28 Jan 2019

Thank you for your valuable comments. To protect the motor of the sampler, the aerosol sampler was stopped for one hour every five hours. Hence, the actual sampling time per membrane was 20 hours. In ultra-sound extraction, the use of 1M HCl was to obtain the maximum amount of nutrients dissolution compared with aerosol water-soluble nutrient. After acidic extraction and filtration, filtrates were adjusted to neutral used

NaOH before the Molybdosilicate Blue methods for P and Si analyses. Fe-P in spr-NW aerosol accounting for 33%, which was higher than Fe-P in its source, Asian sand-dust (0.12%-14%, Yang, 2012), so except for source, Fe-P pattern also probably linked to the modification in atmospheric transport path (e.g. acid processing). In order to avoid errors between experimental methods, only one-site comparison was made. Considered the wider implications for reader's form outside the region, key and generalizable conclusions will be strengthened. The discussion on nitrate/ammonium ratio without Na+ was the most simplified case under the observation that Na+ was not dominant component in water-soluble components of aerosols over the Yellow sea (Yang & Xiu, 2009).

---

## Short Comment (SC5) · 29 Jan 2019

Thank you for your suggestions and comments. First, this island site can ensure the continuity of the sampling in time scale as much as possible, which the cruises over the Yellow sea cannot. Though there were no replicates for sampling sites, the replicate measurement of sub-samples has been done. The relative standard deviation of concentration of nutrients for the replicates (n=5) were less than 2%. Second, so far,

only the comparison between fiber filter and quartz filter on nutrient sampling and analysis (Pszenny et al., 1993) has been reported and there are no related reports about the comparison of the polycarbonate filters and cellulose fiber filters, it acquiesced that there is no difference between them on aerosol bulk sampling for nutrients analysis. None sole use of filter was an unavoidable pity because of the shortage of polycarbonate film in the market and recommendation of cellulose fiber film in national standard. Third, the overall RSD of replicates (n=3) of the dissolution experiment (Milli-Q water as leaching solution, 60min) were for 1.7%, 1.2%, 2.3% and 3.4% ammonium, nitrate phosphate and silicate, respectively. Hence, it was considered that there is no need to repeat.
* * *
[Figure]

[Figure]

Figure. The high-time resolution dissolution curves for replicates (n=3) using raw nutrient concentration data. The leaching solution was Milli-Q water (pH=5.5) and leaching time was 60 min.

**Fig. 1.** The high-time resolution dissolution curves for replicates

---

## Short Comment (SC6) · 30 Jan 2019

Thank you very much for your comment. First, this island site can ensure the continuity of the sampling in time scale as much as possible, which the cruises over the Yellow sea cannot. Second, the reason for the chosen six aerosols was that they captured seasonal dominant aerosol source direction and had more centralized back-trajectories. As for sampling information, it has been partially given in supplementary materials

[Figure]

(Table S1) and it will be completely shown in the table below. Though there were no replicates for the high time-resolution dissolution experiment, the replicate measurement of sub-samples has proved the stability of the dissolution experiment, which the RSD of replicates (n=3) of the dissolution experiment (Milli-Q water as leaching solution, 60min) were for 1.7%, 1.2%, 2.3% and 3.4% for ammonium, nitrate, phosphate and silicate, respectively and detailed information is given in figure below. In addition, the general pattern, such as first-order dissolution reaction of short-time dynamic dissolution of aerosols in seawater, will not change despite of sampling limitations. As for literature writing, such as citation format, word selection, I will be strengthened afterward.

Table. Sampling date, source, TSP mass concentration (μg·m$^{-3}$), mean air pressure (hPa), temperature (℃), wind direction (°) wind speed (m/s) and relative humidity (%) of six aerosols.

| Sample | Date | Source | TSP | Air pressure | Temperature | Wind direction | Wind speed | Relative humidity |
|--------|------|--------|-----|--------------|-------------|----------------|------------|-------------------|
| spr-SW | 2011/4/28 | SW | 35.7 | 1002.8 | 11.8 | 184 | 5.4 | 75 |
| spr-NW | 2011/3/20 | NW | 236.4 | 1011.4 | 7.6 | 325 | 4.5 | 68 |
| sum-NW | 2012/6/12 | NW | 84.1 | 994.9 | 21.8 | 281 | 8.0 | 84 |
| atu-NW | 2012/10/13 | NW | 80.4 | 1011.2 | 20.5 | 309 | 9.4 | 60 |
| win-NW1 | 2011/2/12 | NW | 56.5 | 1019.8 | -2.2 | 338 | 4.2 | 70 |
| win-NW2 | 2011/1/28 | NW | 129.3 | 1025.0 | -1.6 | 314 | 5.6 | 52 |

**Fig. 1.** Sampling information

[Figure]

Figure. The high-time resolution dissolution curves for replicates (n=3) using raw nutrient concentration data. The leaching solution was Milli-Q water (pH=5.5) and leaching time was 60 min.

**Fig. 2.** The high-time resolution dissolution curves for replicates

---

## Short Comment (SC7) · 30 Jan 2019

In fact, the polycarbonate filter has a different pore size from the cellulose fiber filters. This would significantly affect the particle size and composition of aerosol samples as well as nutrients dissolution kinetics of aerosols. Therefore, using two different kinds of filters in collecting TSP samples in the present research is a serious methodological deficiency because it could not guarantee the reliability and comparability of data. No

matter how good the results and discussions are, they cannot make up for this defect.

---

## Short Comment (SC8) · 31 Jan 2019

The ploy-carbonate filter and cellulose filter have different pore size, which were 0.4 $\mu$m and 20 $\mu$m, respectively. The uniform pore size distribution of the ploy-carbonate filter and its light weight seem more appropriate for aerosol collection and dissolution study. The consistency of the use of filter for observation was surely important, however, the filter factory stopped making ploy-carbonate filters within the promotion of cellulose

filters. To continue the observation, we had to choose to change the filters.

If the change of the filter had aerosol collection difference, it would cause less collection efficiency after the filter change theoretically, which would arouse the absolute amount difference directly. However, our results focused more on the relative amount. For example, the comparsion of nutrients between water-soluble and acid-soluble ratio in the ultra-sound extraction and high time-resolution dissolution experiment. Also, dynamic dissolution parameters, such as the dissolution equilibrium time, dissolution constant and the order of the dissolution reaction were not affected by the change of filter. As for dissolution rate comparsion with P and Si minerals, the absolute amount was used. P and Si mainly exist in coarse particles and both filters can capture the coarse particles. Hence, the flaw in aerosol collection did not affect the main conclusions.

---

## Short Comment (SC9) · 31 Jan 2019

As the author stated in the response, the polycarbonate filter and cellulose filter have different pore size, which were 0.4 $\mu$m and 20 $\mu$m, respectively. Obviously, these two kinds of filters have a very big difference in the pore size. This means that the aerosol samples collected using the polycarbonate filters contained a higher proportion of fine particles whereas those collected using the cellulose filters contained a higher propor-

tion of coarse particles due to missing 0.4-20 $\mu$m particulates (this part of fine particles is very important). Please note that the composition and ratio of nutrients in fine particles are absolutely different from those in coarse particles. Therefore, data obtained using two kinds of different filters have no comparability. Once again, using two different kinds of filters in collecting TSP samples in this work is a serious methodological deficiency and thus there is no sense in making further argument.

"However, our results focused more on the relative amount. For example, the comparison of nutrients between water-soluble and acid-soluble ratio in the ultra-sound extraction and high time-resolution dissolution experiment."

The relative amount will be changed owing to missing 0.4-20 $\mu$m particulates.

"Also, dynamic dissolution parameters, such as the dissolution equilibrium time, dissolution constant and the order of the dissolution reaction were not affected by the change of filter."

That is wrong. The results will be significantly altered by the change of filter, because dynamic dissolution parameters differs largely between the fine and coarse particles.

"As for dissolution rate comparsion with P and Si minerals, the absolute amount was used. P and Si mainly exist in coarse particles and both filters can capture the coarse particles. Hence, the flaw in aerosol collection did not affect the main conclusions."

That is not correct. The absolute amount was also affected owing to missing 0.4-20 $\mu$m particulates. P and Si also exist in fines particles, and hence the main conclusions will be influenced by this methodological deficiency.
* * *

---

## Short Comment (SC10) · 31 Jan 2019

As the authors replied, there were no replicates for the high time-resolution dissolution experiment in this study. This is a serious issue the readership will concern. Without data reproducibility experiments, it is difficult to guarantee the reliability of the results.

Moreover, Qianliyan island ($36°16'N$, $120°23'E$) is a very small islet with an area of 0.2 km2 (length: 0.82 km; width: 024 km). It is only one sampling point and cannot

represent the whole Yellow Sea and its coastal areas. A broader sampling area should be covered.

Additionally, in this work the authors selected only six aerosols (spr-SW, spr-NW, sum-NW, aut-NW, win-NW1 and win-NW2) to capture seasonal main features. The number of samples was far too small to study seasonal/annual variation characteristics. Atmospheric Chemistry and Physics journal should publish a large scale study with a wide range of sampling fields and more sample quantity, but not this limited and local study.

---

## Author Comment (AC1) · 14 Mar 2019

Thanks all referees for valuable questions and detail suggestions. Based on all your comments, a detail point-by-point response represented all authors are given as follows.

Response to Anonymous Referee #1

[Figure]

1.Comments from Referee: First, the aerosol samples were collected from only one island site and there were no replicates for sampling sites.

Author's response: Just as Jeju island is a representative site in the East Yellow Sea, Qianliyan island is a representative site in the West Yellow Sea, which on the Asia-Pacific atmospheric mass transport path. Besides, island-based investigation has its advantage over the voyage investigation, ensured the continuity of the weekly sampling for all-year observations, which the cruises over the Yellow sea cannot. The un-executed replicates collection at sampling sites was due to in-sufficient power supply on the island.

Author's changes in manuscript: Add the explanation in 2.1 Site Description, Sample Collection and Sample Selection in manuscript.

2.Comments from Referee: Second, total suspended particulate (TSP) samples were collected by using two different filters in 2011 and 2012 (the poly-carbonate filters in 2011 and Whatman cellulose fiber filters in 2012). I am wondering if these different filters could affect the particle size and composition of the aerosol samples. This should be addressed by an experimental approach.

Author's response: The pore size of the poly-carbonated and the Whatman41 cellulose fiber filter were 0.4$\mu$m and 20$\mu$m, which may probably cause the missing part of fine particles from atmospheric total suspended particles on Whatman41 filter. However, literatures have reported that Whatman41 filter is suitable for high-volume sampling (Fitzgerald & Detwiler, 1955; Fitzgerald & Detwiler, 1957; Lindeken et al., 1963; Watts et al., 1987; Kitto & Anderson,1988). Particularly for fine particles (submicron aerosols), the collection efficiency was over 75% (Fitzgerald & Detwiler, 1955; Fitzgerald & Detwiler, 1957; Lindeken et al., 1963). Besides, experimental results showed that filter efficiency increase rapidly from 75% to 95% within 25min when aerosol diameter was around 0.4$\mu$m and mass concentration was 0.5mgÂům-3 (Lindeken et al., 1963), the filter efficiency was observed to increase rapidly with time. Though the 0.5mgÂům-

3 is higher than usually encountered in atmospheric monitoring, considered that our sampling time was 20h, coarse particles surely blocked the filter pores, narrowed the pore size during long collection period and our sample TSP mass concentration is higher than usually encountered in atmospheric monitoring, thus our collection efficiency for fine particles was over 95%. Therefore, it has minor filter collection efficiency difference between two filters.

Author's changes in manuscript: Add the explanation in supplementary materials.

3.Comments from Referee: Third, the authors collected 39 aerosol samples but they only measured nutrient dissolution from six aerosol samples. I understand that the authors intended to analyze samples by season and main source. However, it is a pity that the nutrient dissolution experiment has no replicates for each category of samples (spr-SW, spr-NW, sum-NW, aut-NW, win-NW1 and win-NW2). This hinders comparison between sample categories by a statistical analysis.

Author's response: Replicates experiments have been done, which the relative standard deviation (RSD) of concentration of nutrients for the ultra-sound replicates (n=5) were less than 2% and that of high time-resolution dissolution replicates (n=3) (Milli-Q water as leaching solution, 60min) were 1.7%, 1.2%, 2.3% and 3.4% for ammonium, nitrate, phosphate and silicate, respectively. The replicates result for each time point during leaching processing was shown in Figure 1 (in this reply).

Author's changes in manuscript: Add the replicates results in 2.2. Ultra-sound Extraction and Chemical Analysis and 2.3. High Time-resolution Dissolution Experiment in manuscript.

4.Comments from Referee: Moreover, the manuscript also requires extensive editing for English language including grammar and word choice.

Author's response: English writing ability needs to be improved by professional native English speaker.

Author's changes in manuscript: Modify English grammar and word choice throughout manuscript.

5.Comments from Referee: Overall, I don't think this manuscript should be considered for publication at this moment.

Author's response: I hope that my changes will satisfy you.

Author's changes in manuscript: Many changes in manuscript.

6.Comments from Referee: L1 Title: The current title is not appropriate for an ACP submission. It needs to be shortened.

Author's response: I agree that the title need to be shorten.

Author's changes in manuscript: Change the title as Aerosol High Time-resolution Nutrient Dissolution Kinetic, Potential Linkages with Inorganic Compositions and P solubility-controlled factors.

7.Comments from Referee: L26-28: It is likely a speculation that is not well supported by current results.

Author's response: "Compared with the slow dissolution of inorganic P and Si, the rapid dissolution of inorganic N can affect the composition of marine nutrients and marine primary productivity," is not well-expressed. The original intention of this sentence was to express that aerosol, as external source of marine nutrients (N, P and Si), its deposition affect the composition of marine nutrients and marine primary productivity, which is evidence-based speculation. Previous research found that N and P are the main factors driving high level productivity in the Yellow Sea (Gao, 2009; Dou et al., 2011). Aerosol soluble N:P is higher than Redfield ratio, which representing the marine biomasses N: P absorption ratio and the surface water N: P, which representing the marine condition, which correspondingly resulting in N increasement in seawater is significant than P increasement in seawater. Considered that aerosol inorganic N dissolution time (within minutes) and P dissolution time (within hours), aerosol particle

residence time is enough for their soluble phase dissolution. Hence, aerosol deposition affects the composition of marine nutrients and marine primary productivity.

Author's changes in manuscript: Revise this sentence in abstract in manuscript.

8.Comments from Referee: L101-118 (Figure 1): Please give more information on the calculation of backward trajectories, including input data, methodology and validation.

Author's response: The airborne 72h backward trajectory in the intervals of an hour at the height of 1000 m, the lower height of marine atmospheric boundary layer (MABL). All backward trajectories were computed by the Hysplist4 software, which has a wide use in atmospheric transport modeling (Wang et al., 2010; Tošić & Unkašević, 2013; Cohen et al., 2015; Chai et al., 2017). The vertical motion method chose input model data, which were downloaded from NOAA FTP server (arlftp.arlhq.noaa.gov/archives).

Author's changes in manuscript: Provide more information on backward trajectory mapping.

9.Comments from Referee: Figures, 3, 5-8: No replicates and no statistical comparison between aerosol sample categories.

Author's response: Though no replicates for six aerosols, replicate experiments for the ultra-sound and the high time-resolution dissolution experiment have been done. These replicate results basically show better parallelism.

Author's changes in manuscript: No change in manuscript.

Response to Anonymous Referee #2

1.Comments from Referee: The authors discuss the detailed comparisons of a few aerosols from this one site, but most readers will be more interested in the generalities of the results which I think can be summarized as inorganic N dissolves completely and rapidly, while only a percentage of P and Si dissolves and that dissolution takes place over timescales of a few hours.

Author's response: I agree that more general results should be provided to readers and accepted the referee's proposal.

Author's changes in manuscript: Adjust the descriptive result of the high-time resolution dissolution experiments in Abstract, 3.3. Depictions and Equations for Dissolution Curves and 5. Conclusion in manuscript.

2.Comments from Referee: I would also suggest that the authors need to note a few caveats of this type of experiment. Firstly, wet deposition dominates in most places. The pH of aerosol or rain depositing to seawater will rise to close to 8 almost immediately so prolonged acid exposures can happen in clouds but will then rapidly be reversed. The timescales of dissolution relevant to marine ecosystems are the lifetimes in the surface mixed layer of particles which are days to weeks so the dissolution rates of even the Si and P specie are rapid with respect to that.

Author's response: These caveats should be properly announced in the manuscript.

Author's changes in manuscript: Add the explanation in 2.3. High Time-resolution Dissolution Experiment in manuscript.

3.Comments from Referee: Line 100 Explain why weekly collections only span 20 hours, I assume it is collecting for 1 day each week.

Author's response: The plan sampling time was intended to be one day; however, the actual sampling time was 20 hours per membrane. The aerosol sampler was stopped for one hour every five hours to protect the motor of the sampler and to avoid unstable voltage on the island as well.

Author's changes in manuscript: Add the explanation in 2.1. Site Description, Sample Collection and Sample Selection in the manuscript.

4.Comments from Referee: Line 120 why was 1M HCl used, that is surely very much more acidic. In addition, the P and Si analyses methods are sensitive to the pH of the analyzed solution, did this cause any issues?

Author's response: In ultra-sound extraction, the use of 1M HCl was to obtain the maximum amount of nutrients dissolution compared with aerosol water-soluble nutrient. After acidic extraction and filtration, filtrates were adjusted to neutral used NaOH before the Molybdosilicate Blue methods for P and Si analyses.

Author's changes in manuscript: Add the explanation of the use of 1M HCl in manuscript.

5.Comments from Referee: Line 226 Why do you link the Fe-P pattern to acid processing rather than source, it seems to me it could be either.

Author's response: Fe-P in spr-NW aerosol accounting for 33%, which was higher than Fe-P in its source, Asian sand-dust (0.12%-14%, Yang, Guo & Li, 2012), so except for source, Fe-P pattern also probably linked to the modification in atmospheric transport path (e.g. acid processing).

Author's changes in manuscript: No change in manuscript.

6.Comments from Referee: Line 280-5 and 328-358 These are quite long discussion sections that could be shortened to focus on key and generalizable conclusions rather than specific comparisons of a few aerosols from this one site – consider the wider implications for reader from outside the region.

Author's response: I accept that Line 280-5 and Line 328-358 should be refined.

Author's changes in manuscript: Rewrite 3.3. Depictions and Equations for Dissolution Curves, 3.4. P solubility and 4.1. Relevance of Aerosol Inorganic Components and Dissolution Patterns.

7.Comments from Referee: The issue of comparisons of rates of dissolution for pure minerals, particularly silicates, to the observed rates (around line 380) are an interesting observation that should be retained.

Author's response: I agree to keep the dissolution rate comparison between aerosol

and pure mineral.

Author's changes in manuscript: No change in manuscript.

8.Comments from Referee: Line 307 The logic of the discussion around nitrate/ammonium ratios seems to me to be a bit flawed. At the very least it ignores sodium nitrate formed by the sea-salt displacement reaction in the coarse mode aerosol. The nitrate/ammonium ratio in an aerosol depends on emission rates and deposition so I'm not sure this section is particularly useful.

Author's response: The discussion on nitrate/ammonium ratio without Na+ was the most simplified case based on the observation that Na+ was not dominant component in water-soluble components of aerosols along the coastal zone of the Shandong Peninsula, which overlapped the Qianliyan island (Yang & Xiu, 2009). The discussion around nitrate/ammonium ratio have defects in adapting to actual aerosols and its low necessity to the key and general conclusion, therefore this part cannot appear on manusecript.

Author's changes in manuscript: Delete the discussion nitrate/ammonium ratio content.

Response to Williams

1.Comments from Referee: In this study, a series of nutrient dissolution experiments were conducted to determine the soluble fraction of atmospheric nutrients and revealed the short-time dissolution processes, patterns and kinetics of nutrient elements in aerosols. However, I found that the method used to collect aerosol samples had a significant drawback. Total suspended particulate (TSP) samples were collected using poly-carbonate filters in 2011 and Whatman cellulose fiber filters in 2012. Why were different filters used in collecting TSP samples in 2011 and 2012? Was the influence of different filters assessed before the use? This is an important issue because it will largely impact nutrients dissolution kinetics of aerosols and their controlling factors. Supposing both filters are suitable for TSP collection and chemical analysis, but the

authors should have used the same kind of filter in the same sampling site for the collection of different season samples. This is essential to maintain the reliability and comparability of the data.

Author's response: The change of filter was a helpless move due to no poly-carbonate filter provider at that sampling period. To keep up the sampling, Whatman41 film had to be used. The pore sizes of the poly-carbonated and the Whatman41 cellulose fiber filter were 0.4$\mu$m and 20$\mu$m, which might arise fine particles loss on Whatman41 filter. However, former literatures have reported that Whatman41 filter is suitable for high-volume sampling (Fitzgerald & Detwiler, 1955; Fitzgerald & Detwiler, 1957; Lindeken et al., 1963; Watts et al., 1987; Kitto & Anderson,1988). Particularly for fine particles (submicron aerosols), the collection efficiency was over 75% (Fitzgerald & Detwiler, 1955; Fitzgerald & Detwiler, 1957; Lindeken et al., 1963). Besides, experimental results showed that filter efficiency increase rapidly from 75% to 95% within 25min when aerosol diameter was around 0.4$\mu$m and mass concentration was 0.5mgÂům-3 (Lindeken et al., 1963), the filter efficiency was observed to increase rapidly with time. Though the 0.5mgÂům-3 is higher than usually encountered in atmospheric monitoring, considered that our sampling time was 20h, coarse particles surely blocked the filter pores, narrowed the pore size during long collection period and our sample TSP mass concentration is higher than usually encountered in atmospheric monitoring, thus our collection efficiency for fine particles was over 95%. Therefore, it has minor filter collection efficiency difference between two filters.

Author's changes in manuscript: Add the information to further explain that filter replacement had less impact.

2.Comments from Referee: Moreover, the sample number 6 in this study is too small to interpret temporal change of aerosol dissolution processes. More than that, I did not find the detailed processes and mechanism on nutrients elements dissolution in aerosols in the present study. The authors should carefully address this issue in more details.

[Figure]

Author's response: Selection of 39 samples was aimed to analyze samples by season and main source. Though sample size was small, they basically captured the seasonal characteristics of the main source direction and satisfied the aim. In this research, the high time-resolution dissolution experiment provides the detailed processes, which cannot get from the most of the total extraction experiments; P form correlated with P dissolution parameters and aerosol P and Si dissolution rates compared with pure minerals provided the potential mechanism of the bulk aerosol, rather than microscopic level mechanism. With the help of sing particle collection, high-resolution electron microscope and other well-developed techniques, more mysteries of microscopic dissolution will be revealed in the near future.

Author's changes in manuscript: Add explanation for choosing six samples.

3.Comments from Referee: Overall, this study is a local investigation in a small island but not focused on studies with general implications for atmospheric science. I do not feel that this manuscript fits the scope of Atmospheric Chemistry and Physics due to its too local interest and its defect in sample collection methods.

Author's response: The Qianliyan island is a representative site in the West Yellow Sea, just as Jeju island is a representative site in the East Yellow Sea, which both on the Asia-Pacific atmospheric mass transport path. It implies that aerosols at this location are both from Asia representing the land and/or the Pacific Ocean representing ocean, which covered large source area. Considered that source effect on aerosol nutrient characteristics (Arnold et al., 1998), the study on aerosol properties at this island has universality in nearby areas. Besides, the issue filter change was explained at first.

Author's changes in manuscript: Add the information on the importance of study area in manuscript.

Response to Zhang

1.Comments from Referee: After reading this manuscript, I fįnd that there are serious

issues in this study. The Qianliyan Island was only one very small sampling site.

Author's response: the Qianliyan island is a representative site in the West Yellow Sea, just as Jeju island is a representative site in the East Yellow Sea, which on the Asia-Pacific atmospheric mass transport path. It implies that aerosols at this location are both from Asia representing the land and/or the Pacific Ocean representing ocean, which covered large source area.

Author's changes in manuscript: Add the information on the importance of study area in manuscript.

2.Comments from Referee: In more marked deficiencies, only 6 aerosol samples were collected during a short period at this small island. Research area is most important for atmospheric science, but I have no idea from this manuscript. Atmospheric Chemistry and Physics (ACP) should publish a large scale study with a wide range of sampling fields (e.g., covering the whole Yellow Sea and its coastal areas) and enough samples (e.g., at least 12 months samples which covered a whole year). Therefore, this limited research is not suitable to be published in ACP.

Author's response: Selection from 39 samples was aimed to analyze samples by season and main source, of which seasonal main source can get from daily back trajectories during the collection period. Though sample size was small, they basically represented the seasonal main source/direction characteristics and close backward trajectories for each collection period, which satisfying the aim. Also, the leaching curves of the six aerosols which illustrate that although their absolute concentrations and solubilities are different, their dissolution equilibrium time (different element dissolution time is different, same element dissolution time is similar) and the similar dissolution pattern (dissolution rate goes from fast to slow) has already offered the uniformity rule of aerosol dissolution. From both the ultra-sound extraction and the leaching dissolution experiments, they concluded similar rules for acid stimulation of the same element. In summary, the above experiments meet the origin purpose.

Author's changes in manuscript: Add explanation for choosing six samples and highlight the importance of study area in manuscript.

3.Comments from Referee: These 6 aerosol samples capturing seasonal dominant aerosol sources were selected to carry out the high time-resolution dissolution experiments. It was not clear why only six samples were selected. The 6 aerosol samples are indeed too few to study temporal variation. Moreover, the significance of this research site was not described in detail.

Author's response: The reason for choosing six samples to do the high time-resolution dissolution experiments was to fit the aim of selection that maximized to capture seasonality features of aerosols to the utmost based on the frequency of four directions of seasonal daily back trajectories. The daily -72h back-trajectory of air-masses in spring, summer, autumn, and winter during 2011-2012 at Qianliyan Island were shown in Figure S2 in supplementary material. As for a single sample, the least alternations in source direction were required among around 20 backward trajectories at 1h intervals. In all, 6 samples avoided drastic changes in aerosol source during sampling. The core of this manuscript was not for temporal variation, but the main dissolution characteristics for Asian-West Pacific (terrestrial-ocean) regional aerosol. The main findings of this manuscript are as follows. From both the ultra-sound extraction and the leaching dissolution experiments, they concluded that similar rules for acid stimulation of the same element. It specifically meant that N dissolution was a fast process; while P and Si dissolution was a rather long process and had potential for further release if aerosol met atmospheric acidification and kept the modified soluble phase. From the leaching curves, although aerosol absolute concentrations and solubilities are different, their dissolution equilibrium time (different element dissolution time is different, same element dissolution time is similar) and the similar dissolution pattern (dissolution rate goes from fast to slow), which offered the uniformity rule of aerosol dissolution.

Author's changes in manuscript: Add explanation for choosing six samples in manuscript.

4.Comments from Referee: Sampling information (including sampling location, date, time etc.) of six samples are vital to data analyze and discussion; unfortunately, they were not described.

Author's response: Aerosol sampling information has been partially given in supplementary materials (Table S1) and it will be completed shown in the Table 1.

Author's changes in manuscript: Give the complete information in supplementary materials.

5.Comments from Referee: All data necessary to understand, evaluate, replicate, and build upon the reported research must be made available and accessible whenever possible.

Author's response: All data results from experiments were described, analyzed and concluded. As for replicate, we did replicated experiments for each experiment. The RSD of all nutrient concentrations for the ultra-sound replicates (n=5) were less than 2% and that of high time-resolution dissolution replicates (n=3) (Milli-Q water as leaching solution, 60min) were 1.7%, 1.2%, 2.3% and 3.4% for ammonium, nitrate, phosphate and silicate, respectively. (The replicates result for each time point during leaching processing was shown in Figure 1 (in this reply).) As for data availability, the data of this paper are available upon request (sumeiliu@ouc.edu.cn).

Author's changes in manuscript: Add replicate experiment information in manuscript.

6.Comments from Referee: Critically, I do not see any reference to the air pressure, temperature, wind direction, wind speed, relative humidity, TSP, nutrients concentrations data, not even in the table.

Author's response: Aerosol sampling information (the air pressure, temperature, wind direction, wind speed, relative humidity and TSP) are given in the Table 1 (in this reply) and nutrients concentrations data are given in Figure 3, 4, 5 and 7 in the manuscript.

Author's changes in manuscript: Give the complete information in supplementary materials.

7.Comments from Referee: Lines 615-623: Three references cite as Chen et al. 2006 were shown. If two or more references from the same year contain the same first six or more authors, use a, b, c, and so on for the in-text citation and in the references list. Please check ACP reference format.

Author's response: It's my mistake and I will correct it immediately.

Author's changes in manuscript: Correct related citations and references.

8.Comments from Referee: It is noted that this manuscript needs careful editing by someone with expertise in technical English editing paying particular attention to English grammar, spelling, and sentence structure so that the goals and results of the study are clear to the reader.

Author's response: Poor English editing will be strengthened afterward.

Author's changes in manuscript: Ask professional native English speaker to correct language mistakes in literature.

References:

Arnold, E., Merrill, J., Leinen, M., & King, J. (1998). The effect of source area and atmospheric transport on mineral aerosol collected over the North Pacific Ocean. Global and Planetary Change, 18(3–4), 137–159. http://doi.org/10.1016/S0921-8181(98)00013-7.

Chai, T., Crawford, A., Stunder, B., Pavolonis, M. J., Draxler, R., & Stein, A. (2017). Improving volcanic ash predictions with the HYSPLIT dispersion model by assimilating MODIS satellite retrievals. Atmospheric Chemistry and Physics, 17(4), 2865-2879. http://doi.org/10.5194/acp-17-2865-2017.

Cohen, M. D., Ngan, F., Rolph, G. D., Draxler, R. R., Stunder, B. J. B., & Stein, A. F. (2015). NOAA's HYSPLIT Atmospheric Transport and Dispersion Modeling System. Bulletin of the American Meteorological Society, 96(12), 2059–2077. http://doi.org/10.1175/bams-d-14-00110.1.

Dou, Y., Tang, X., Yang, Z., & Wang, Y. (2011). Study on the effects of nutrient structure on primary productivity in the Yellow Sea of Shandong offshore. Marine Environmental Science, 177-181. (in Chinese version)

Fitzgerald, J. J., & Detwiler, C. G. (1955). Collection efficiency of air cleaning and air sampling filter media. American Industrial Hygiene Association Quarterly, 16 (2), 123-130. http://doi.org/10.1080/00968205509344466.

Fitzgerald, J. J., & Detwiler, C. G. (1957). Collection efficiency of filter media in the particle size range of 0.005 to 0.1 micron. American Industrial Hygiene Association Quarterly, 18 (1), 47-54. http://doi.org/10.1080/00968205709343462.

Gao, S. (2009). Spatial and temporal variations of chlorophyll and primary productivity in the North Yellow Sea and their influencing factors. (Master dissertation, China Ocean University; in Chinese version)

Kitto, M. E., & Anderson, D. L. (1988). The use of whatman41 filters for particle. Atmospheric Environment, 22(11), 2629-2630. http://doi.org/10.1016/0004-6981(88)90500-8.

Lindeken, C. L., Morgin, R. L., & Petrock, K. F. (1963). Collection efficiency of Whatman 41 filter paper for submicron aerosols. Health Physics, 9, 305-308.

Tošić, I., & Unkašević, M. (2013). Extreme daily precipitation in Belgrade and their links with the prevailing directions of the air trajectories. Theoretical and Applied Climatology, 111(1–2), 97–107. http://doi.org/10.1007/s00704-012-0647-5.

Wang, F., Chen, D. S., Cheng, S. Y., Li, J. B., Li, M. J., & Ren, Z. H. (2010). Identification of regional atmospheric PM10 transport pathways using HYSPLIT, MM5-CMAQ and synoptic pressure pattern analysis. Environmental Modelling and Software, 25(8), 927–934. http://doi.org/10.1016/j.envsoft.2010.02.004.

Watts, S. F., Yaaqubi, R., & Davies, T. (1987). The use of Whatman 41 filter papers for high volume aerosol sampling. Atmospheric Environment, 21(12), 2731-2732. http://doi.org/10.1016/0004-6981(87)90207-1.

Yang, G. P., & Xiu, L. P. (2009). Chemical characteristics of aerosol-bound ionic species over the southern coastal area of Shandong Peninsula. Journal of Ocean University of China (natural science edition), 39(4), 745-749. (in Chinese version)

Yang, H. W., Guo, B. S., & Li, J. W. (2012). Distribution of phosphorus forms in desert and dust particles in Inner Mongolia and their environmental significance. Environmental Chemistry, 31(7). (in Chinese version)
* * *
[Figure]

Figure 1. The high-time resolution dissolution curves for replicates (n=3) using accumulated nutrient concentration (error bar: the standard deviation). The leaching solution was Milli-Q water (pH=5.5) and leaching time was 60 min.

[Figure]

[Figure]

**Fig. 1.** Figure 1. The high-time resolution dissolution curves for replicates

Table 1. Sampling date, source, TSP mass concentration (µg·m$^{-3}$), mean air pressure (hPa), temperature (°C), wind direction (°) wind speed (m/s) and relative humidity (%) of six aerosols.

| Sample | Date | Source | TSP | Air pressure | Temperature | Wind direction | Wind speed | Relative humidity |
|--------|------|--------|-----|--------------|-------------|----------------|------------|-------------------|
| spr-SW | 2011/4/28 | SW | 35.7 | 1002.8 | 11.8 | 184 | 5.4 | 75 |
| spr-NW | 2011/3/20 | NW | 236.4 | 1011.4 | 7.6 | 325 | 4.5 | 68 |
| sum-NW | 2012/6/12 | NW | 84.1 | 994.9 | 21.8 | 281 | 8.0 | 84 |
| atu-NW | 2012/10/13 | NW | 80.4 | 1011.2 | 20.5 | 309 | 9.4 | 60 |
| win-NW1 | 2011/2/12 | NW | 56.5 | 1019.8 | -2.2 | 338 | 4.2 | 70 |
| win-NW2 | 2011/1/28 | NW | 129.3 | 1025.0 | -1.6 | 314 | 5.6 | 52 |

**Fig. 2.** Table 1. Sampling information

---

## Editor Comment (EC1) · Chak K. Chan (Editor) · 15 Mar 2019

After reviewing the responses to all the comments from reviewers and others, I feel that the paper has still a lot of improvements to be made before it can adequately address these major concerns on sample size, data analysis, and the development of the major conclusions. I decide that the paper is not suitable for further review.